# Recent Research Progress on Natural Stilbenes in *Dendrobium* Species

**DOI:** 10.3390/molecules27217233

**Published:** 2022-10-25

**Authors:** Denghui Zhai, Xiaofa Lv, Jingmei Chen, Minwen Peng, Jinyan Cai

**Affiliations:** 1Key Laboratory of Glucolipid Metabolic Disorder of Ministry of Education of China, Key Unit of Modulating Liver to Treat Hyperlipemia SATCM, Guangdong Metabolic Disease Research Center of Integrated Chinese and Western Medicine, Guangdong Pharmaceutical University, Guangzhou 510006, China; 2Center for Drug Research and Development, Guangdong Pharmaceutical University, Guangzhou 510006, China

**Keywords:** *Dendrobium*, stilbenes, phenanthrenes, bibenzyls, antioxidant, anti-inflammatory, antitumor, anti-α-glucosidase inhibitory activities

## Abstract

*Dendrobium* is the second biggest genus in the Orchidaceae family, and many of them have been utilized as a traditional Chinese medicine (TCM) for thousands of years in China. In the last few decades, constituents with great chemical diversity were isolated from *Dendrobium*, and a wide range of biological activities were detected, either for crude extracts or for pure compounds. Stilbene compound is one of the primary active constituents in the genus *Dendrobium*. At present, 267 stilbene compounds with clarified molecular structures have been extracted and isolated from 52 species of *Dendrobium*, including 124 phenanthrenes and 143 bibenzyls. At the same time, activity studies have indicated that 157 compounds have pharmaceutical activity. Among them, most of the compounds showed antitumor activity, followed by antioxidant, anti-inflammatory and anti-α-glucosidase inhibitory activities. Additionally, 54 compounds have multiple pharmacological activities, such as confusarin (14), 2,4,7-trihydroxy-9,10-dihydro-phenanthrene (43), moscatilin (148), gigantol (150) and batatasin III (151). This review summarizes current knowledge about the chemical composition of stilbene, bioactivities and pharmacologic effects in 52 species of *Dendrobium*. We also expect to provide a reference for further research, development and utilization of stilbene constituents in the *Dendrobium* genus.

## 1. Introduction

The genus *Dendrobium*, which belongs to the family Orchidaceae, includes more than 1500 species widely distributed throughout Asia, Europe and Australia [1]. According to the literature, there are almost 80 species of this genus in China; many of them are used in traditional or folk medicine [2]. *Dendrobium* serves numerous functions in traditional Chinese medicine, including as an antipyretic, as an anti-inflammatory agent, for benefit of the eyes and digestive system, and regulating blood sugar level [3]. Recent scientific research on the genus led to the isolation of a series of compounds, such as stilbenes, polysaccharides, alkaloids, amino acids, sesquiterpenes, fluorenones, flavonoids, phenolic acids, phenylpropanoids, lignans, amides, alkaloids and steroids [4]. Pharmacological studies have disclosed that some of them display wide bioactive activities.

The stilbene components mainly include phenanthrenes and bibenzyls in *Dendrobium* species. The variety of the substituents on the benzene ring endows stilbene compounds with numerous biological activities. In recent years, more studies have focused on the stilbene components of *Dendrobium* and multiple pharmacological effects have been discovered. Preclinical studies have found that the phenanthrene compounds in *Dendrobium* species showed good antioxidant, anti-inflammatory and antitumor activity [5]. Meanwhile, bibenzyls, another class of stilbene compounds in *Dendrobium* species, have attracted attention as well. At present, nearly 150 bibenzyls have been extracted and tested from *Dendrobium* species, some of which showed various pharmacological activities, such as antitumor, anti-inflammatory, anti-diabetes and its complications [5]. Although stilbene compounds are considered to constitute a relatively small group of natural products, stilbenes have good development prospects for its various monomer components, broad pharmacological effects and a wide range of sources; therefore, discovering new stilbene constituents and evaluating their prospective biological activities have become of great interest to many research groups worldwide. We collected and analyzed *Dendrobium* species used in traditional medicines or ethnomedicine, now a well-studied species, and reviewed the pharmacological activity and mechanisms of the stilbene components in 52 *Dendrobium* species. “*Dendrobium*”, “stilbenes”, “phenanthrenes” and “bibenzyls” were used as search terms to screen the literature. Cited references were collected between 1995 and 2022 from the Web of Science, China National Knowledge Internet (CNKI), SciFinder, Google Scholar, Baidu Scholar, Pubmed and GeenMedical. All studies were independently screened, and the included data were reviewed, analyzed and summarized.

## 2. Phenanthrenes

Phenanthrene is an important chemical component of stilbene in *Dendrobium* species. The phenanthrene compounds from the *Dendrobium* have a variety of structures with a different substitution number and position of the methoxy and the hydroxyl group on the aromatic ring. Typically, phenanthrenes can be divided into five types, including simple phenanthrene, dihydrophenanthrene, phenanthraquinone, diphenanthrene and phenanthrene derivatives. In our analysis of 52 species of *Dendrobium*, only 44 species of *Dendrobium* were found to contain phenanthrene, except *D. capillipes*, *D. catenatum*, *D. findlayanum*, *D. huoshanense*, *D. harveyanum*, *D. hercoglossum*, *D. secundum* and *D. williamsonii*. Currently, 124 phenanthrene compounds have been identified and isolated from 44 species of *Dendrobium*. Among these species, *D. nobile* has the maximum number of phenanthrene, containing 37 monomer components, followed by *D. denneanum*, containing 16 monomer components. The distribution and structure of these compounds in *Dendrobium* are shown in Table 1 and Figure 1.

### 2.1. Simple Phenanthrene

The structural characteristics of simple phenanthrene are generally only hydroxy and methoxy substituents on the aromatic ring. Their great structural diversity stems from the number and position of their oxygen functions. The hydroxy and methoxy moieties numbers are between 3 and 6, and can usually be found on C-2, C-3, C-4, C-5, C-6 or C-7. At present, 31 simple phenanthrene compounds, **1–31**, have been extracted and isolated from 44 species of *Dendrobium*. Among these species, moscatin (1) is the most widely distributed simple phenanthrene, with an amount of 14, followed by nudol (15), with an amount of 8. In addition, substitutions at position C-1, C-8, C-9 and C-10 are quite rare: compounds **23**, **24** and **25** substituted with a methoxy group at C-1; compounds **9**, **14** and **17** substituted with a methoxy group at C-8; compound 26 substituted with a methoxy group at C-9; compound **27** substituted with a hydroxy group at C-9; and compound **28** substituted with a methoxy group at C-9 and C-10.

### 2.2. Dihydrophenanthrene

The main difference between dihydrophenanthrene and simple phenanthrene is that the C-9 and C-10 sites are linked by single bonds. Dihydrophenanthrene substituents are also hydroxyl and methoxy, generally located at positions 2, 3, 4, 5 and 7, which are basically consistent with phenanthrene compounds. Currently, 36 dihydrophenanthrene compounds, **32–67**, have been extracted and isolated from 44 species of *Dendrobium*. Meanwhile, substitutions at position C-1, C-6, C-8 and C-9 are quite rare: compound **65** substituted with a methoxymethyl group at C-1; compounds **66** and **67** substituted with a hydroxy group at C-1; compounds **34**, **57** and **60** substituted with a methoxy group at C-6; compound **62** substituted with a methoxy group at C-8; and seven compounds **35–41** substituted with a methoxy group at C-9. Among the 44 species of *Dendrobium*, hircinol (53) is the most widely distributed dihydrophenanthrene, with an amount of 15, followed by the coelonin (55), with an amount of 9.

### 2.3. Other Phenanthrenes

A total of 16 phenanthraquinone (68–83) were isolated from 19 species of *Dendrobium*. Among these species, *D. nobile* has the maximum number, containing seven monomer compounds, followed by *D. wardianum*, containing three monomer components, and other species containing minor monomer compounds with a number ranging from 1 to 2. In addition, phenanthrene-1,4-quinones are the most commonly occurring phenanthrenequinones, densiflorol B (68) is the most widely distributed phenanthraquinone in our analysis of the *Dendrobium* species, with an amount of 6. A total of 14 diphenanthrene (84–97) were isolated from *D. nobile*, *D. plicatile*, *D. palpebrae*, *D. senile*, *D. thyrsiflorum* and *D. venustum*. In addition, *D. nobile* isolated the maximum number, containing seven monomer components, *D. plicatile* and *D. palpebrae* isolated the minimum number, containing one monomer compound. Most diphenanthrene are synphenes connected by C-C, with the main positions of 1-1’ connections. A total of 27 phenanthrene derivatives (98–124) were isolated from *D. loddigesii*, *D. chrysotoxum*, *D. chrysanthum*, *D. denneanum*, *D. fimbriatum*, *D. nobile*, *D. officinale*, *D. signatum* and *D. parishii*. Meanwhile, *D. denneanum* isolated the maximum number, containing eleven monomer components, followed by *D. loddigesii*, containing six monomer compounds.

## 3. Bibenzyls

Bibenzyls are another class of stilbene chemical ingredients that have been widely reported in the genus *Dendrobium*. Bibenzyl compounds refer to compounds formed by two benzyl units linked by a single methyl C-C bond. The character of bibenzyls is that the C3, C5 and C-4′ positions are usually hydroxyl or methoxy on the core structure, and the C2 or/and C4 often have a p-hydroxyl or phenyl substitution. Although the structure of the parent nucleus of bibenzyl compounds is simple, the position and number of methoxy and hydroxyl groups on the aromatic ring, the bridge chain substituents connecting the aromatic ring and the change of the aromatic ring leads to various structural types and, thus, to various biologic activities. Bibenzyls can be divided into simple bibenzyl, bridged carbon bibenzyl and bibenzyl derivatives. In our analysis of 52 species of *Dendrobium*, only *D. hainanense* did not discover bibenzyl; the other species all had different amounts of bibenzyl. Currently, 143 bibenzyl compounds (**125–267**) have been discovered and isolated from 51 species of *Dendrobium*. Among these species, *D. nobile* has the maximum number of bibenzyl, containing 32 monomer components, followed by *D. candidum* and *D. gratiosissimum*, containing 29 and 21 monomer components. The distribution and structure of these compounds in *Dendrobium* are shown in Table 2 and Figure 2.

### 3.1. Simple Bibenzyl

Currently, 68 simple bibenzyl (**125–192**) have been discovered and separated from 51 species of *Dendrobium*. The structural characteristics of simple bibenzyl are generally only hydroxy and methoxy substituents on the aromatic ring. Regardless of the simplicity of the skeletons of simple bibenzyl compounds, they have various structural types due to the position and number of methoxy and hydroxyl groups on the aromatic ring. The hydroxy and methoxy substituents numbers are between 3 and 6, and can usually be found on C-3, C-4, C-5, C-3′, C-4′ or C-5′. In addition, substitution at position C-2 is quite rare: compound 181 substituted with a hydroxy group at C-2 and compound 190 substituted with a methoxy group at C-2. Moscatilin (148) and gigantol (150) were found in most *Dendrobium*, with the species amount of 35 and 30, respectively. Batatasin III (151), tristin (130) and 4,4′-dihydroxy-3,5-dimethoxybibenzyl (157) were also found in *Dendrobium*, with the species amounts of 23, 16 and 10, respectively.

### 3.2. Bridged Carbon Bibenzyl

The main difference between bridged carbon bibenzyl and simple bibenzyl lies in the change of the aromatic ring and the bridge chain substituents connecting the aromatic ring. Similarly, bridged carbon bibenzyl has various structural types due to the position and number of methoxy and hydroxyl groups on the aromatic ring. In addition, it also exhibits various structural types depending on the substituents of the bridge chain. The number of hydroxyl and methoxy substituents on the aromatic rings ranges from 4 to 5, and can normally be found on C-3, C-4, C-5, C-3′, C-4′ or C-5′. Up to now, 22 bridged carbon bibenzyl (193–214) have been discovered and isolated from 14 species of *Dendrobium*, including *D. aphyllum*, *D. candidum*, *D. crepidatum*, *D. heterocarpum*, *D. wardianum*, *D. fimbriatum*, *D. findlayanum*, *D. hancockii*, *D. officinale*, *D. hercoglossum*, *D. nobile*, *D. lindleyi*, *D. loddigesii* and *D. sinense*. Among these species, *D. nobile* has the maximum number of bridged carbon bibenzyl, containing six monomer components, followed by *D. candidum* and *D. hercoglossum*, containing three monomer components.

### 3.3. Bibenzyl Derivatives

The genus *Dendrobium* is one of the primary sources of bibenzyl derivatives. Although, the structure complexity of bibenzyl derivatives means that multiple structural types are also due to the position and number of methoxy and hydroxyl groups on the aromatic ring, and changes in the aromatic ring. At present, 53 bibenzyl derivatives (215–267) have been identified and isolated from 27 species of *Dendrobium*. Among these species, *D. candidum* has the maximum number of bibenzyl derivatives, containing fourteen monomer components, followed by *D. fimbriatum*, containing nine monomer components and other species containing minor monomer components, with amounts ranging from 1 to 7.

## 4. Pharmacological Activity of Stilbenes in *Dendrobium* Species

Of the 267 isolated stilbene constituents, 157 compounds were found to have pharmacological activities, 103 monomer compounds showed single pharmacological activities and 54 monomer compounds had multiple pharmacological activities. In the following, stilbene constituents that were isolated from the different *Dendrobium* species are described according to different pharmacological effects, such as anti-oxidant, anti-inflammatory, antitumor, α-Glucosidase and pancreatic lipase inhibitory activities, such as antimicrobial, neuroprotective and anti-platelet aggregation effects.

### 4.1. Anti-Oxidant Activity

Oxidative stress is believed to be implicated in aging, neurodegenerative disease, diabetes mellitus, cardiovascular disease and tumors. Antioxidants can be profitable in combating these processes and diseases. The DPPH and ABTS free radical have been widely used to evaluate the antioxidant capacity of fractions and pure compounds to obtain the more effective and nontoxic compounds for the prevention and/or treatment of such diseases [77]. There are 58 stilbene constituents in the genus *Dendrobium*, which showed antioxidant activity by DPPH and ABTS free radicals scavenging assays (Table 3). While positive results have been obtained for the in vitro anti-oxidant activity of most stilbene components, further human studies are needed to substantiate the beneficial and negative effects of these compounds.

In isoamoenylin (127), a bibenzyl isolated from the *D. amoenum*, the antioxidative activity was determined by the nitro blue tetrazolium (NBT) method, which showed moderate superoxide-scavenging activity (IC_50_:694 µM) [130]. Flavanthrinin (2), dendroparishiol (121), dendrocandin E (145), moscatilin (148), 4,3′,4′-trihydroxy-3,5-dimethoxybibenzyl (160) and 4,5,4′-trihydroxy-3,3′-dimethoxy-bibenzyl (161) were isolated from the whole plant of *D. parishii*. These compounds were evaluated for their free radical scavenging activities using three methods, including ORAC, DPPH and deoxyribose assays. For each assay, each compound was initially tested at a concentration of 50 μg/mL. The results showed that compound 121 had the highest antioxidant activity in all chemical assays. Other compounds showed moderately potent antioxidant activities. Accordingly, dendroparishiol (121) was selected for further evaluation of antioxidant by induced oxidative stress in RAW264.7 cells with H_2_O_2_. The results showed that compound 121 diminished intracellular ROS in RAW264.7 cells by 65% at the maximum concentration of 50μg/mL, and enhanced antioxidant enzyme (SOD, GPx and CAT) activities in H_2_O_2_ treated RAW264.7 cells [71].

Many compounds have been discovered to have excellent in vitro antioxidant activity; however, their in vivo activities and mechanisms of action have rarely been explored. In 2021, su, et al. [131] assessed the protective effect of lusianthridin (30) on hemin-induced low-density lipoprotein oxidation (he-oxLDL). The research demonstrated that compound 30 protected LDL oxidation induced by hemin, and had the potential protective effect in foam cell formation. In addition, compound 30 could inhibit the formation of TBARs, reduce REM, reduce oxidized lipid products, as well as preserve the level of cholesterol arachidonate and cholesterol linoleate (Figure 3).
molecules-27-07233-t003_Table 3Table 3Antioxidant activity of stilbene compounds isolated from *Dendrobium* genus.No.Compound*Dendrobium* SpeciesDPPH Radical ScavengingABTS Radical ScavengingRef.1**1***D. loddigesii*62.2 µM-[52]2**2***D. nobile*IC_50_: 35.71 ± 0.19 µM-[61]3**7***D. nobile*IC_50_: 29.67 ± 1.11 µM-[61]4**14***D. nobile*IC_50_: 12.90 ± 0.35 µM-[61]5**26***D. nobile*IC_50_: 34.78 ± 0.04 µM-[61]6**27***D. loddigesii*IC_50_: 26.1 µM-[52]7**31***D. nobile*IC_50_: 144.5 ± 2.65 µM-[61]8**43***D. loddigesii*IC_50_: 14.1 µM-[51]9**46***D. draconis*IC_50_: 11.7 µM-[36]10**53***D. draconis*IC_50_: 283.3 ± 13.7 µM-[36]11**73***D. draconis*IC_50_: 22.3 ± 1.0 µM-[36]12**79***D. aphyllum*5.25%-[9]13**99***D. loddigesii*IC_50_: 23.2 µM-[51]14**126***D. signatum*IC_50_: 9.91 ± 0.3 µMIC_50_: 12.0 ± 0.3 µM[78]15**130***D. chrysanthum*IC_50_: 14.70 µM-[24]16**132***D. williamsonii*IC_50_: 19.5 µM-[132]17**133***D. loddigesii*IC_50_: 85.8 µM-[51]18**135***D. heterocarpum*0.68 ± 1.45%31.13 ± 3.19%[43]19**140***D. heterocarpum*33.13 ± 2.20%28.32 ± 2.94%[43]20**145***D. candidum*IC_50_: 15.6 µM-[92]21**147***D. nobile*IC_50_: 40.3 ± 0.1µM-[115]22**148***D. nobile*IC_50_: 14.5 ± 0.3 µM-[115]23**149***D. nobile*IC_50_: 14.0 ± 0.1 µM-[115]24**150***D. heterocarpum*5.36 ± 1.09%40.97 ± 6.55%[43]25**151***D. heterocarpum*1.93 ± 1.24%34.88 ± 4.11%[43]26**157***D. signatum*IC_50_: 14.0 ± 0.1 µMIC_50_: 18.5 ± 0.2 µM[78]27**161***D. secundum*IC_50_: 15.87 ± 1.48 µM-[132]28**166***D. loddigesii*89.411%-[50]29**167***D. catenatum*IC_50_: 34.45 ± 1.07 µMIC_50_: 9.01 ± 1.39 µM[97]30**185***D. nobile*IC_50_: 21.8 ± 0.4 µM-[115]31**191***D. loddigesii*29.292%-[50]32**192***D. loddigesii*12.041%-[50]33**196***D. heterocarpum*21.79 ± 1.01%16.58 ± 2.75%[43]34**197***D. nobile*IC_50_: 19.9 ± 0.8 µM-[115]35**198***D. candidum*IC_50_: 34.2 µM-[92]36**199***D. candidum*IC_50_: 34.5 µM-[92]37**208***D. loddigesii*35.276%-[50]38**215***D. aphyllum*87.97%-[9]39**216***D. loddigesii*IC_50_: 60.1 µM-[51]40**218***D. heterocarpum*21.08 ± 1.19%29.89 ± 5.38%[43]41**219***D. candidum*IC_50_: 32.4 µM-[93]42**220***D. candidum*IC_50_: 36.8 µM-[94]43**221***D. candidum*IC_50_: 70.2 µM-[94]44**223***D. nobile*IC_50_: 21.0 ± 0.4 µM-[115]45**224***D. signatum*IC_50_: 8.9 ± 0.1 µMIC_50_: 18.0 ± 0.1 µM[78]46**225***D. signatum*IC_50_: 15.6 ± 0.2 µMIC_50_: 8.1 ± 0.0 µM[78]47**226***D. candidum*IC_50_: 19.8 µM-[93]48**227***D. candidum*IC_50_: 45.0 µM-[94]49**229***D. candidum*IC_50_: 87.6 µM-[94]50**230***D. candidum*IC_50_: 50.4 µM-[94]51**231***D. catenatum*-IC_50_: 10.03 ± 0.88 µM[97]52**232***D. candidum*IC_50_: 40.5 µM-[94]53**233***D. candidum*IC_50_: 22.3 µM-[94]54**234***D. candidum*IC_50_: 30.3 µM-[94]55**237***D. findlayanum*37.563 ± 0.099%-[106]56**255***D. findlayanum*18.115 ± 0.478%-[106]57**256***D. findlayanum*27.632 ± 0.347%-[106]58**257***D. signatum*IC_50_: 10.2 ± 0.1 µMIC_50_: 24.0 ± 0.2 µM[78]


### 4.2. Anti-Infammatory Activity

Macrophages play a major role in inflammation and host defense mechanisms against bacterial and viral infections. The NO radical, synthesized by the oxidation of L-arginine catalyzed by nitric oxide synthase, is involved in a number of physiological and pathological processes in mammals. Meanwhile, excessive production of NO by iNOS in macrophages is involved in various acute and chronic inflammatory diseases. Therefore, inhibitors of NO production in macrophages are an important target in the treatment of certain inflammatory diseases [59]. Due to some phenolic compounds from different *Dendrobium* species showing significant inhibitory effects on NO production, discovering new phenolic compounds and evaluating their prospective anti-inflammatory activities have become of great interest to many research groups worldwide. At present, 45 stilbene constituents in the genus *Dendrobium* have shown anti-inflammatory activity in LPS-stimulated RAW264.7 cells by MTT assay (Table 4). Among them, compounds **1**, **32**, **37**, **38**, **43**, **52**, **55**, **58**, **63**, **98**, **99**, **103**, **107**, **115**, **116**, **133**, **148**, **197** and **223** exhibited significant anti-inflammatory activity, while other compounds showed moderately potent anti-inflammatory activity.

Some compounds with anti-inflammatory activity have been explored for mechanisms. The anti-inflammatory activities of dehydroorchino (11) and ephemeranthol A (63) were discovered to be caused by a blockage of NF-κB activation and phosphorylation of MAP kinases in the macrophages [63]. Furthermore, 5-methoxy-2,4,7,9S-tetrahydroxy-9,10-dihydro-phenanthrene (37) and 2,5-dihydroxy-4-methoxy-phenanthrene 2-O-β-_D_-glucopyranoside (103) inhibited NO production by blocking iNOS expression, p38 MAPK phosphorylation and IκBα phosphorylation [29]. Dendrochrysanene (101), which has been confirmed to suppress the mRNA level of TNF-α, IL8, IL10, and iNOS in LPS-stimulated mouse peritoneal macrophages [22]. Dendroparishiol (121) decreased NO and TNF-α secretion, and reduced iNOS and COX-2 expression in a dose dependent manner from LPS treated RAW264.7 cells [71]. Gigantol (150) alleviated liver inflammation partly through inhibiting the JNK/cPLA2/12-LOX pathway [133]. In vivo, crepidatin (185) exhibited a significant protective effect against LPS-induced acute lung injury in mice [100]. Furthermore, 3′,4-dihydroxy-3,5′-dimethoxybibenzyl (189) remarkably reduced NF-κB, IκB, ERK, JNK, p38 and Akt phosphorylation of LPS-induced RAW264.7 macrophages [109]. Additionally, the mRNA levels of iNOS, TNF-α and IL-1β induced by LPS could also be remarkably inhibited by compounds 136, 139, 224 and 231, in a dose-independent manner (Figure 3) [108].

### 4.3. α-. Glucosidase and Pancreatic Lipase Inhibitory Activities

α-Glucosidase is a membrane bound enzyme in the small intestine, which is responsible for digesting starch and disaccharide into glucose. Inhibition of this enzyme can delay carbohydrate digestion, which can prevent excess glucose absorption. Pancreatic lipase is the key enzyme responsible for lipid digestion. Inhibition of pancreatic lipase enzyme can reduce lipid absorption and may help to protect pancreas β-cells [47]. In recent years, the number of reports on natural compounds with α-glucosidase and lipase inhibitory activities have continuously increased. Significant research has been focused on the search for alternative α-glucosidase inhibitors with non-sugar core structure, particularly the polyphenols, due to their abundant availability in nature and their promising biological activities [41]. At present, 29 stilbene constituents in the genus *Dendrobium* showed α-glucosidase and pancreatic lipase inhibitory activities (Table 5).

The research found that the inhibition of carbohydrate and lipid hydrolyzing enzymes by compound 73 may decrease the rate of their cleavage, component release and absorption in the small intestine, and consequently, suppress postprandial hyperglycemia and hyperlipidemia. Additionally, lusianthridin (32) and moscatilin (148) at non-toxic concentrations may find a role in helping to regulate glucose metabolism by stimulating glucose uptake in skeletal muscle, the largest site of glucose disposal, leading to the prevention of MetS and type II diabetes [41]. In addition, dendrofalconerol B (263), a bisbibenzyl compound, exhibited strong pancreatic lipase and α-glucosidase effects. This compound also exhibited anti-adipogenic activity through the suppression of PPARγ and C/EBPα expression (Figure 3) [111].

### 4.4. Antitumor Activity

There are 70 stilbene compounds in the genus *Dendrobium* that have exhibited antitumor activity. Among them, 19 compounds were active against single cancer cells and 51 compounds were active against multiple cancer cells (Table 6 and Table 7). Moscatilin (148) is the most common stilbene compound of the genus *Dendrobium*, which also has anticancer activity against a variety of cancer cells. Moscatilin was shown to induce apoptosis in human colorectal cancer cells through tubulin depolymerization, DNA damage and c-Jun N-terminal kinase (JNK) activation [134]; apoptosis of human pancreatic cancer cells via reactive oxygen species and the JNK/stress-activated protein kinases (SAPK) pathway [135]; and apoptosis and mitotic catastrophe in human esophageal cancer cells by early promotion of the M phase cell cycle blockade and the regulation of mitotic catastrophe-associated proteins [136]. Moscatilin induces apoptosis of human pharyngeal squamous carcinoma cells by activating caspases through the JNK signaling pathway [137]. Moscatilin impedes HCC invasion and uPA expression through the Akt/NF-κB signaling pathway [138]. Moscatilin was also reported to inhibit the migration and metastasis of human breast cancer cells by inhibiting Akt and the Twist signaling pathway [139]. In addition, moscatilin suppressed tumor angiogenesis and growth in human umbilical vein endothelial cells, blocking ERK1/2, Akt and the eNOS pathway [140]. 

Studies on the anti-cancer mechanism of other stilbenes have found that nudol (15) caused cell cycle arrest at G2/M phase in U2OS cells, and it also induced cell apoptosis through the caspase-dependent pathway and inhibited migration of OS cells [141], and chrysotoxene (23)-induced apoptosis of HepG2 cells in vitro and in vivo via activation of the mitochondria-mediated apoptotic signaling pathway [142]. Lusianthridin (32) suppresses lung cancer stem cells via the inhibition of Src-STAT3-c-Myc; thereby, regulating the stemness of the cells [143]. The research disclosed that cypripedin (69) possesses cytotoxic activity against NSCLC and synergies cisplatin-induced cell death. The underlying mechanism is believed to be through the downregulation of the anti-apoptotic protein Bcl-xL, which results in the imbalance of apoptosis regulatory proteins, causing the loss of mitochondrial membrane integrity and the release of cytochrome c [144]. Denbinobin (71) has been reported to inhibit A549 cells and human glioblastoma multiforme cell apoptosis through caspase and apoptosis signal-regulating kinase 1 activation. In addition, compound 71 inhibits CXCL12-induced PC3 cell migration by inhibiting Rac1 activity [145]. phoyunnanin E (94) inhibits migration and growth in an anchorage-independent manner with detailed mechanisms of action covering EMT suppression, reduction of migratory-associated integrins αv and β3 and suppression of FAK/Akt signals which consequently suppressed downstream migratory proteins in lung cancer [146]. In addition, Phoyunnanin-E-mediated apoptosis via a p53-dependent pathway by increasing the accumulation of cellular p53 protein [147]. Chrysotoxine (149) markedly suppressed 6-OHDA-induced apoptosis in human neuroblastoma SH-SY5Y cells, presumably through direct scavenging of intracellular ROS induced by 6-OHDA, thus blocking ROS-mediated downstream signaling pathways [148]. In addition, compound **149** was shown to suppress active Src/Akt signal and in turn depleted Sox2-mediated cancer stem cells [149]. Batatasin III (151) has promising anti-cancer properties by inhibiting cell proliferation, migration and invasion by suppressing EMT and FAK/AKT/CDC42 pathways [150]. Chrysotobibenzyl (172) inhibited lung cancer cell migration via Cav-1, integrins β1, β3 and αν, and EMT suppressions [151]. Alteration to down-stream signaling of p53 including activation of pro-apoptosis protein, reduction of anti-apoptosis and suppression on protein kinase B (Akt) survival pathway were notified in 4,5,4′-trihydroxy-3,3′-dimethoxybibenzyl (161)-treated lung cancer cells [152]. Erianin (184) down-regulates the expression of inflammation factors through the regulation of IDO-induced tumor cells angiogenesis mimicry and endothelial cell-dependent angiogenesis by targeting the JAK2/STAT3 pathway and its down-stream genes MMP-2/MMP-9, and thereby inhibits the angiogenesis of lung cancer cells (Figure 3) [153].

### 4.5. Antimicrobial Activity

In isoamoenylin (127), a bibenzyl isolated from *D. amoenum*, the antibacterial activity was determined by the agar cup-plate diffusion method. The results indicated that the zone of inhibition (diameter in mm) at a concentration of 200 µg/mL against *Pseudomonas aeruginosa*, *Escherichia coli*, *Bacillus subtilis* and *Staphyllus aures* was 8.5, 9.0, 8.5 and 8.5 mM, respectively [130]. Tristin (130), 3,3′,5-trihydroxybibenzyl (133), 3-hydroxy-5-methoxybibenzyl (139), batatasin III (151) and 3′,4-dihydroxy-3,5′-dimethoxybibenzyl (189) were isolated from the stems of *D. nobile*. These compounds were evaluated against the phytopathogenic fungi *Alternaria brassicicola*, *Phytophthora parasitica* var. *nicotianae*, *Colletotrichum capsici*, *Bipolaris oryzae*, *Diaporthe medusaea nitschke*, *Ceratocystis paradoxa moreau*, *Exserohilum turcicum*, *Pestallozzia theae* and *Alternaria citri*. The results indicated that these compounds exhibited broad-spectrum antifungal activity against these phytopathogenic fungi. Compounds 130, 133, 139 and 151 exhibited especially antifungal activity against *Bipolaris oryzae*, with MIC values ranging from 96.2 to 109.6 µM, compared to 133.3 µM for prochloraz. Compound 189 exhibited antifungal activity against *Diaporthe medusaea nitschke*, with an MIC value of 91.2 µM, compared to 133.3 µM for prochloraz [56]. In 2012, Chen et al. isolated a new phenanthrenequinone, named denbinobin B (76) from the whole plant of *D. sinense*. The compound was tested for antibacterial activity in vitro against SA strains by the filter paper disk agar diffusion method. Antibacterial assay exhibited that denbinobin B (76) (20.0 mg/mL) showed an inhibitory effect on SA with the diameter of inhibition zone 16.5 mM [81].

### 4.6. Neuroprotective Activity

ACh (acetyl-choline) and BCh (butyrylcholine) are required for cholinergic neurotransmission in the central and peripheral nervous systems. AChE and BChE activity have been used as markers for cholinergic activity, which plays a crucial role in learning and memory. Chrysotoxine (149), chrysotobibenzyl (172) and erianin (184) were isolated from the stems of *D. chrysotoxum*, and were also assessed for inhibitory activity of two enzymes, AChE and BChE. The results showed that compounds 149, 172 and 184 had a certain inhibitory activity on BChE with the inhibition (%) 19.35, 30.68 and 41.66, respectively. Meanwhile, compounds 149 and 184 had weak inhibitory activity on AChE, with the inhibition (%) 9.87 and 14.48 [98]. Furthermore, 3,7-dihydroxy-2,4-dimethoxy-phenanthrene (7), nudol (15), 3,4,7-trihydroxy-2-methoxy-phenanthrene (17), 3-hydroxy-2,4,7-trimethoxy-9,10-dihydrophenanthrene (54) and erianthridin (58) were isolated from the aerial parts of *D. hainanense*. These compounds were evaluated for inhibitory activity of acetylcholinesterase. The results showed that compounds 7, 15, 17, 54 and 58 had a certain inhibitory activity on AChE with the inhibition (%) 25.73, 18.11, 19.90, 14.94 and 17.96, respectively [45]. Furthermore, 2,7-dihydroxy-3,4,6-trimethoxy-9,10-dihydrophenanthrene (60) was isolated from the whole plant of *D. sinense*, and evaluated against for inhibitory activity of acetylcholinesterase. The results showed that compound 60 had a certain inhibitory activity on AChE with the inhibition (%) 1.98 [82]. Confusarin (14) and crepidatuol A (242) were isolated from the stems of *D. crepidatum*, and evaluated for their enhancing activity on NGF-induced neurite outgrowth in PC12 cells. The results indicated that the proportion of the NGF-induced (10 μg/mL) neurite-bearing cells were enhanced by confusarin (7.1%) and crepidatuol A (8.5%), respectively, at the concentration of 10 µM [27].

### 4.7. Anti-Platelet Aggregation Activity

Moscatilin (148) and gigantol (150) were isolated from the stems of *D. densiflorum*, and evaluated for their anti-platelet aggregation activity in vitro. As a result, moscatilin and gigantol exhibited anti-platelet aggregation activity on SD rat platelet aggregation in preliminary pharmacological tests in vitro [33]. Four compounds were isolated from the stems of *D. longicornu*, including 7-methoxy-9,10-dihydrophenanthrene-2,4,5-triol (46), 5-methoxy-7-hydroxy-9,10-dihy-dro-1,4-phenanthrenequinone (73), aloifol I (128) and longicornuol A (244). These compounds were tested in vitro for anti-platelet aggregation activity on New Zealand white rabbit platelets. The results indicated that these compounds all had weak anti-platelet aggregation activities [54]. Trigonopol A (245) was isolated from the stems of *D. trigonopus*, and tested for anti-platelet aggregation activity on New Zealand white rabbit platelets in vitro. Trigonopol A exhibited moderate antiplatelet aggregation activity in vitro with 67.55% inhibitory ration at 1.4337 × 10^−3^ M [87].

### 4.8. Others

Nine bibenzyls were isolated from the whole plant of *D. findlayanum*, including 3,4’-dihydroxy-5-methoxybibenzyl (135), 3,3’-dihydroxy-5-methoxybibenzyl (137), 3,4,4′-trihydroxy-5-methoxybibenzyl (141), 3,4-dihydroxy-3’,4’,5-trimethoxybibenzyl (142), 4,4′-dihydroxy-3,5-dimethoxybibenzyl (157), 3′,4-dihydroxy-3,5-dimethoxybibenzyl (158), 4,4′-dihydroxy-3,5,3′-trimethoxybibenzyl (171), 3,3′-dihydroxy-4,5′-dimethoxybiphezyl (187) and (R)-3,α-dihydroxy-4,4ʹ,5-trimethoxybibenzyl (207). These compounds were assessed for their activity in promoting the gastrointestinal motility of zebrafish treated with Nile red. Strikingly, all the compounds, except 135 and 207, could promote gastrointestinal motility at the Nile red excretion at doses of 8 µM. In particular, compound 171 had significant activity to promote the gastrointestinal motility of zebrafish at the concentration of 0.3 µM [105].

Ten bibenzyls were isolated form the stems of *D. loddigesii*, including tristin (130), 3,3′,5-trihydroxybibenzyl (133), moscatilin (148), batatasin III (151), 4,5,4′-trihydroxy-3,3′-dimethoxybibenzyl (161), 4′,5-dihydroxy-3,3′-dimethoxybiphezyl (166), crepidatin (185), aphyllal C (191), densiflorol A (192) and (R)-4,5,4ʹ-trihydroxy-3,3ʹ,α-trimethoxybibenzyl (208). These compounds were evaluated for their tyrosinase inhibitory activity and their effects on collagen production in HDFa. In tyrosinase inhibitory assay, 3,3′,5-trihydroxybibenzyl (133) exhibited significant inhibitory activity with IC_50_ 37.904 μg/mL. Aphyllal C (191) showed moderate inhibition with IC_50_:152.56 μg/mL. Other compounds were lacking in activities. In collagen production by HDFa assay, batatasin III (151) showed significant stimulation of HDFa collagen production activity with EC50:3.182 μg/mL. Compounds 161 and 166 showed weaker activities, with collagen production of 29.157% and 33.062% at 10 μg/mL, respectively. Other compounds, except 208, showed extremely weak activities with collagen production ranging from 10.871% to 24.552% at 10 μg/mL [50].

## 5. Future Prospects

Stilbenes and their derivatives have attracted increasing attention due to their diverse biological activities and potential pharmacological applications. Probably the most extensively investigated compound is resveratrol, for which more than 2000 papers have been published. The stilbene constituents isolated from the genus *Dendrobium* have also shown di-verse activities, including anti-oxidant, anti-inflammatory, antitumor, anti-α-glucosidase inhibitory, antimicrobial, neuroprotective and anti-platelet aggregation activities, and we consider that some of these have the potential to be developed as drug candidates. The most promising ones are the anti-oxidant (compounds **14**, **26**, **43**, **46**, **99**, **126**, **130**, **145**, **148**, **149**, **161**, **166**, **185**, **197**, **214**, **223**–**226**, **233**), anti-inflammatory (compounds **1**, **32**, **37**, **38**, **43**, **52**, **55**, **58**, **63**, **98**, **99**, **103**, **107**, **115**, **116**, **133**, **148**, **197**, **223**), anti-α-glucosidase inhibitory (compounds **33**, **73**, **115**, **116**, **135**, **144**, **150**, **151**, **263**, **264**) and antitumor (compounds **26**, **32**, **71**, **77**, **94**, **148**–**151**, **157**, **249**–**251**, **255**, **258**) effects. Some pharmacological results con-firm the traditional uses of *Dendrobium* plants, but almost all data is derived from in vitro investigations. Therefore, to certify the effectiveness of these stilbene compounds in hu-man therapy, further investigations, especially in vivo and human studies, are needed.

After nearly two decades of research, significant progress has been made in isolation and structure identification, and a series of naturally occurring stilbenes with various structures have been isolated. However, activity investigations on most compounds are limited to in vitro screening, and few systematic pharmacological studies are available in the literature due to the scarcity of samples. The synthesis of active stilbenes must be developed to obtain sufficient samples for systematic pharmacological studies. In addition, due to their rapid metabolism and excretion in mammalian bodies, improvements in de-livery systems, stability and solubility are required.

While there is still a long way to go, *Dendrobium* species are undeniably one of the most promising sources of biologically active stilbenes. Meanwhile, hopefully even more research groups will deal with the phytochemistry, pharmacology, bioavailability and potential utilization of *Dendrobium.*

## 6. Conclusions

*Dendrobium* has been used as a medicinal herb in China for thousands of years. Nonetheless, due to the high degree of personalization of TCM diagnosis and treatment of TCM, its clinical efficacy cannot be comprehensively evaluated by evidence-based medicine. The fruitful achievements in its chemical constituent investigations in recent years have laid a foundation for the study of its therapeutically material basis and action mechanism, which will help clarify its diverse pharmacologic effects and develop into a modern pharmaceutical preparation.

Stilbenes are rich sources of lead compounds in the search for new drugs and medi-cines. *Dendrobium* species are promising sources of biologically active stilbenes. At present, 267 stilbene compounds with clarified molecular structures have been extracted and isolated from 52 species of *Dendrobium*, that includes 31 simple phenanthrene, 36 dihydrophenanthrene, 16 phenanthraquinone, 14 diphenanthrene, 27 phenanthrene derivatives, 68 simple bibenzyl, 22 bridged carbon bibenzyl and 53 bibenzyl derivatives. Meanwhile, activity studies have shown that 157 stilbene compounds have pharmaceutical activity. Among them, 62 compounds showed antioxidant activity; 51 compounds showed anti-inflammatory activity; 31 compounds showed anti-α-glucosidase and pancreatic lipase inhibitory activities; 73 compounds exhibited antitumor activity; 7 compounds showed antimicrobial activity; 11 compounds exhibited neuroprotective effects; 7 compounds showed anti-platelet aggregation effects; 7 compounds showed gastrointestinal motility activities and 7 compounds showed tyrosinase inhibitory and anti-aging activities. In addition, 19 phenanthrenes and 35 bibenzyls have multiple pharmacological activities, such as confusarin (14), 2,4,7-trihydroxy-9,10-dihydrophenanthrene (43), tristin (130), 3,3′,5-trihydroxybibenzyl (133), 3,4’-dihydroxy-5-methoxybibenzyl (135), moscatilin (148), gigantol (150), batatasin III (151), 4,4′-dihydroxy-3,5-dimethoxybibenzyl (157) and dendrocandin U (231). These compounds showed a variety of pharmacological activities, mainly including antioxidant, anti-inflammatory, antitumor, anti-α-glucosidase inhibitory activities. For many compounds, however, despite having been discovered to have good inhibitory activity in vitro results, their in vivo activities had rarely been determined. It is hoped that the pharmacological mechanism of these compounds can be further studied in the future. In addition to their activity, the pharmacological toxicity and physicochemical properties of the compounds also deserve attention. Here we briefly review the published studies of the last 20 years that were related to stilbene chemical constituents and pharmacologic activities in 52 species of *Dendrobium*. This can not only provide significant pharmacological effect and chemotaxonomic knowledge of the genus *Dendrobium*, but also provide a scientific basis for developing new medicines utilizing this interesting plant and stilbene constituents.

## Figures and Tables

**Figure 1 molecules-27-07233-f001:**
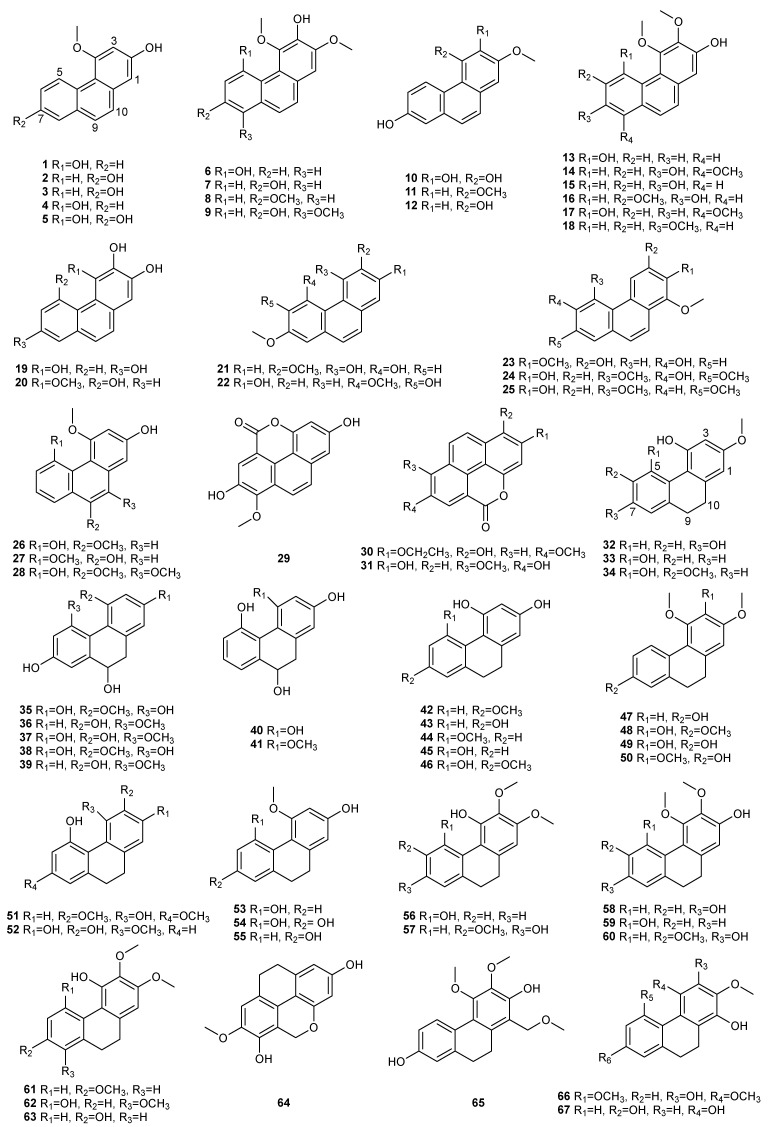
The structures of phenanthrene compounds (**1**–**124**).

**Figure 2 molecules-27-07233-f002:**
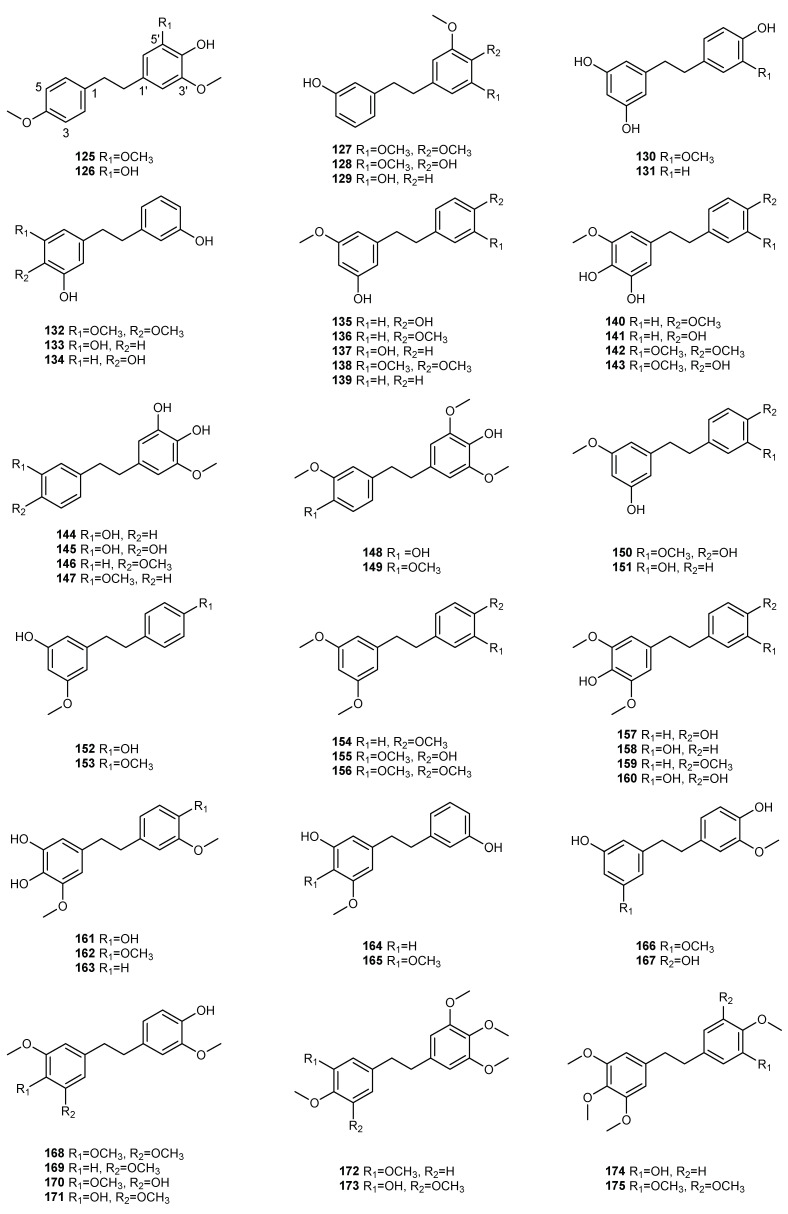
The structures of bibenzyl compounds (**125**–**267**).

**Figure 3 molecules-27-07233-f003:**
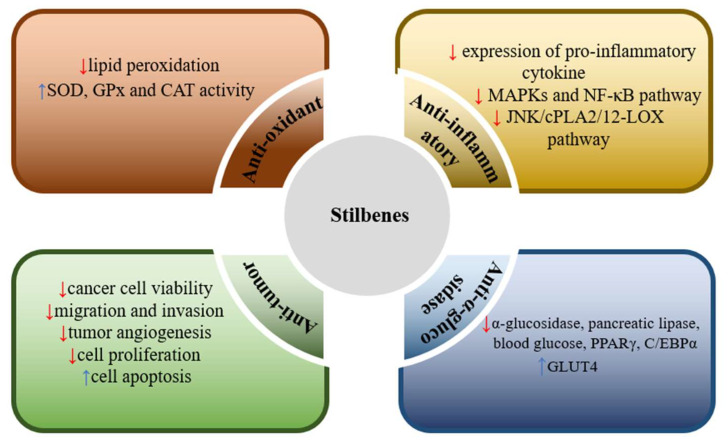
Major biological activities and their mechanisms of action of stilbenes in *Dendrobium* species. Anti-oxidant (inhibiting lipid peroxidation, increasing SOD, GPx and CAT activity); Anti-inflammatory (through inhibition of expression of pro-inflammatory cytokines, such as TNF-α, IL-6, IL-1β, blockage of NF-κB activation and phosphorylation of MAP kinases, inhibiting the JNK/cPLA2/12-LOX pathway); Anti-α-glucosidase (through inhibition α-glucosidase and pancre-atic lipase, suppression of PPARγ and C/EBPα expression); Anti-cancer (through inhibition cancer cell viability, migration and invasion, tumor angiogenesis, cell proliferation and increasing cell apoptosis) activities.

**Table 1 molecules-27-07233-t001:** Phenanthrenes isolated in *Dendrobium* species.

NO.	*Dendrobium* Species	Phenanthrenes	Ref.
1	*D. amoenum*	Flaccidin (64)	[6]
2	*D. aphyllum*	moscatin (1), lusianthridin (32), 4-methoxy-2,5,7,9-tetrahydroxy-9,10-dihydrophenanthrene (35), 5-methoxy-4,7,9-trihydroxy-9,10-dihydrophenanthrene (36), 2,4-dihydroxy-7-methoxy-9,10-dihydro-phenanthrene (42), 2,4,7-trihydroxy-9,10-dihydrophenanthrene (43), hircinol (53), aphyllone A (79)	[7,8,9]
3	*D. bellatulum*	2,5,7-trihydroxy-4-methoxy-9,10-dihydrophenanthrene (54)	[10]
4	*D. brymerianum*	flavanthrinin (2), lusianthridin (32), hircinol (52), densiflorol B (62)	[11,12]
5	*D. candidum*	bulbophyllanthrin (6), 2,5-dihydroxy-3,4-dimethoxyphenanthrene (13), confusarin (14), nudol (15), 2,3,4,7-tetramethoxyphenanthrene (19), 2,4,7-trihydroxy-9,10-dihydrophenanthrene (43), denbinobin (71)	[13,14]
6	*D. capillipes*	-	-
7	*D. catenatum*	-	-
8	*D. christyanum*	4,5-dihydroxy-2-methoxy-9,10-dihydrophenanthrene (33)	[15]
9	*D. chrysotoxum*	moscatin (1), 3,7-dihydroxy-2,4-dimethoxyphenanthrene (7), confusarin (14), nudol (15), 2,7-dihydr-oxy-3,4,6-trimethoxyphenanthrene (16), chrysotoxene (23), 2,5-dihydroxy-4,9-dimethoxyphenanthr-ene (26), 2,7-dihydroxy-8-methoxyphenanthro [4,5-bcd] pyran-5(5H)-one (29), 4,5-dihydroxy-2,6-dimethoxy-9,10-dihydrophenanthrene (34), 2,4,7-trihydroxy-9,10-dihydrophenanthrene (43), eriant-hridin (58), densiflorol B (68), chrysotoxol A (99), chrysotoxol B (100)	[16,17,18,19,20,21]
10	*D. chrysanthum*	moscatin (1), flavanthrinin (2), 2,5-dihydroxy-4,9-dimethoxyphenanthrene (26), loddigesiinol A (27), 2,4-dihydroxy-5-methoxy-9,10-dihydrophenanthrene (44), 2,4,5-trihydroxy-9,10-dihydrophenanthr-ene (45), hircinol (53), coelonin (55), dendrochrysanene (111), denchryside A (112)	[22,23,24,25,26]
11	*D. crepidatum*	confusarin (14), hircinol (53)	[27]
12	*D. crystallinum*	cystalltone (30)	[28]
13	*D. denneanum*	moscatin (1), 4,5-dihydroxy-2-methoxy-9,10-dihydrophenanthrene (33), 5-methoxy-2,4,7,9S-tetrahy-droxy-9,10-dihydrophenanthrene (37), 4-methoxy-2,5,7,9S-tetrahydroxy-9,10-dihydrophenanthrene (38), 5-methoxy-4,7,9S-trihydroxy-9,10-dihydrophenanthrene (39), 2,5-dihydroxy-4-methoxy-phena-nthrene 2-O-β-_D_-glucopyranoside (103), 2,5-dihydroxy-4-methoxy-phenanthrene 2-O-β-_D_-apiofuran-osyl-(1–6)-β-_D_-glucopyranoside (104), 2,5-dihydroxy-4-methoxy-phenanthrene 2-O-α-L-rhamnopyr-anosyl-(1–6)-β-_D_-glucopyranoside (105), 9,10-dihydrophenanthrene 2-O-β-_D_-glucopyranoside (106), 1,2,5,9R-tetrahydroxy-9,10-dihydrophenanthrene 5-O-β-_D_-glucopyranoside (107), denneanoside A (108), denneanoside B (109), denneanoside C (110), denneanoside D (111), denneanoside F (112), denneanoside E (113)	[29,30,31,32]
14	*D. densiflorum*	2,6-dihydroxy-1,5,7-trimethoxyphenanthrene (24), lusianthridin (32), densiflorl B (68), cypripedin (69)	[33]
15	*D. devonianum*	4-methoxy-2,7-dihydroxyphenanthrene (3), 4-methoxy-2,5-dihydroxyphenanthrene (4), 4,5-dihydro-xy-2-methoxy-9,10-dihydrophenanthrene (33), 4,5-dihydroxy-2,6-dimethoxy-9,10-dihydrophenant-hrene (34), 2,4,7-trihydroxy-9,10-dihyrophenanthrene (43), hircinol (53), dendrodevonin A (80), den-drodevonin B (81)	[34,35]
16	*D. draconis*	7-methoxy-9,10-dihydrophenanthrene-2,4,5-triol (46), hircinol (53), 5-methoxy-7-hydroxy-9,10-dihy-dro-1,4-phenanthrenequinone (73)	[36]
17	*D. ellipsophyllum*	4,5-dihydroxy-2,3-dimethoxy-9,10-dihydrophenanthrene (56)	[37]
18	*D. fimbriatum*	moscatin (1), confusarin (14), fimbriatone (31), lusianthridin (32), (S)-2,4,5,9-tetrahydroxy-9,10-dihy-drophenanthrene (40), 2,4,7-trihydroxy-9,10-dihydrophenanthrene (43), 2,4-dimethoxy-9,10-dihy-drophenanthren-7-ol (47), hircinol (53), aphyllone A (79), 9,10-dihydro-aphyllone A-5-O-β-_D_-glucop-yranoside (114)	[38,39,40]
19	*D. findlayanum*	-	-
20	*D. formosum*	confusarin (14), nudol (15), lusianthridin (32), hircinol (53), 2,5,7-trihydroxy-4-methoxy-9,10-dihy-drophenanthrene (54), coelonin (55), erianthridin (58), 5-methoxy-7-hydroxy-9,10-dihydro-1,4-phen-anthrenequinone (73)	[41]
21	*D. gratiosissimum*	flavanthrinin (2), cannithrene 2 (51)	[28]
22	*D. huoshanense*	-	-
23	*D. hancockii*	moscatin (1), 2,5-dihydroxy-4,9-dimethoxyphenanthrene (26), ephemeranthoquinone (74)	[42]
24	*D. harveyanum*	-	-
25	*D. heterocarpum*	coelonin (55)	[43]
26	*D. hercoglossum*	-	-
27	*D. hongdie*	moscatin (1), nudol (15), ephemeranthoquinone (74)	[44]
28	*D. hainanense*	3,7-dihydroxy-2,4-dimethoxyphenanthrene (7), 3-hydroxy-2,4,7-trimethoxy-phenanthrene (8), 3,4,7-trihydroxy-2-methoxyphenanthrene (10), nudol (15), 3-hydroxy-2,4,7-trimethoxy-9,10-dihydro-phenanthrene (48), flavanthridin (49), 7-hydroxy-2,3,4-trimethoxy-9,10-dihydrophenanthrene (50), erianthridin (58), 3,4-dimethoxy-1-(methoxymethyl)-9,10-dihydrophenanthrene-2,7-diol (65)	[45,46]
29	*D. infundibulum*	dendroinfundin A (61), dendroinfundin B (62), ephemeranthol A (63)	[47]
30	*D. lindleyi*	flavanthrinin (2), dehydroorchinol (11), lusianthridin (32), coelonin (55), densiflorol B (68), cypriped-in (69)	[48,49]
31	*D. loddigesii*	moscatin (1), loddigesiinol A (27), lusianthridin (32), plicatol C (41), 2,4,7-trihydroxy-9,10-dihydro-phenanthrene (43), hircinol (53), chrysotoxol A (99), loddigesiinol B (98), loddigesiinol I (115), loddigesiinol J (116), loddigesiinol G (123), loddigesiinol H (124)	[50,51,52,53]
32	*D. longicornu*	7-methoxy-9,10-dihydrophenanthrene-2,4,5-triol (46), hircinol (53), coelonin (55), 5-methoxy-7-hydr-oxy-9,10-dihydro-1,4-phenanthrenequinone (73), ephemeranthoquinone (74)	[54,55]
33	*D. nobile*	moscatin (1), flavanthrinin (2), 3,7-dihydroxy-2,4-dimethoxyphenanthrene (7), denthyrsinin (9), dehydroorchinol (11), lusianthrin (12), confusarin (14), nudol (15), 3,4,8-trimethoxyphenanthrene-2,5-diol (17), fimbriol B (20), 4,5-dihydroxy-3,7-dimethoxyphenanthrene (21), 5,7-dimethoxyphenan-threne-2,6-diol (22), 2,5-dihydroxy-4,9-dimethoxyphenanthrene (26), fimbriaton (31), lusianthridin (32), 4,5-dihydroxy-2-methoxy-9,10-dihydrophenanthrene (33), ephemeranthol C (52), hircinol (53), coelonin (55), erianthridin (58), ephemeranthol A (63), densiflorol B (68), cypripedin (69), 6,7-dihydr-oxy-2-methoxy-1,4-phenanthrenedione (70), denbinobin (71), 7-hydroxy-9,10-dihydro-1,4-phenanth-renedione (75), denobilone B (82), denobilone C (83), denthyrsinol A (84), denthyrsinol B (85), denthy-rsinol C (86), denthyrsinol (89), phochinenin D (93), phochinenin G (95), 4,4′,7,7′-tetrahydroxy-2,2′-dimethoxy-9,9′,10,l0′-tetrahhydro-1,1′-phenanthrene (96), dendronbibisline A (117), dendronbibis-line B (118)	[56,57,58,59,60,61,62,63]
34	*D. moniliforme*	3,7-dihydroxy-2,4-dimethoxyphenanthrene (7), confusarin (14), hircinol (53), 1,5-dihydroxy-3,4,7-trimethoxy-9,10-dihydrophenanthrene (66), denbinobin (71), moniliformin (78)	[64,65,66]
35	*D. officinale*	confusarin (14), nudol (15), 2,4,7-trihydroxy-9,10-dihydrophenanthrene (43), 2,4-dimethoxy-9,10-dihydrophenanthren-7-ol (47), erianthridin (58), ephemeeranthol A (63), dendrocandin P1 (119), den-drocandin P2 (120)	[67,68,69]
36	*D. pachyglossum*	4,5-dihydroxy-2,3-dimethoxy-9,10-dihydrophenanthrene (56)	[70]
37	*D. palpebrae*	2,6-dihydroxy-1,5,7-trimethoxyphenanthrene (24), 2,5-dihydroxy-4,9-dimethoxyphenanthrene (26), lusianthridin (32), dendropalpebrone (97)	[71]
38	*D. parishii*	flavanthrinin (2), dendroparishiol (121).	[72]
39	*D. plicatile*	3,7-dihydroxy-2,4-dimethoxyphenanthrene (7), denthyrsinin (9), nudol (15), plicatol A (28), lusianth-ridin (32), plicatol C (41), flavanthridin (49), hircinol (53), coelonin (55), erianthridin (58), ephemeran-thol A (63), 1,4,7-trihydroxy-2-methoxy-9,10-dihydrophenanthrene (67), ephemeranthoquinone (74), 2,2′-dimethoxy-9,9′,10,l0′-tetrahhydro-1,1′-phenanthrene (96)	[73,74,75]
40	*D. polyanthum*	moscatin (1), hircinol (53), 2,4,7-trihydroxy-9,10-dihydrophenanthrene (43)	[76]
41	*D. pulchellum*	fimbriaton (31)	[77]
42	*D. signatum*	dendrosignatol (122)	[78]
43	*D. scabrilingue*	dendroscabrol A (19), lusianthridin (32), coelonin (55)	[79]
44	*D. secundum*	-	-
45	*D. senile*	moscatin (2), 2,5,7-trihydroxy-4-methoxyphenanthrene (5), 2,5-dihydroxy-4,9-dimethoxyphenanthr-ene (26), bleformin G (90), 4,4′,8,8′-tetramethoxy [1,10-biphenanthrene]-2,2′,7,7′-tetrol (91), 2,2′,7,7′-tetrahydroxy-4,4′-dimethoxy-1,10-biphenanthrene (92)	[80]
46	*D. sinense*	2,5,7-trihydroxy-4-methoxy-9,10-dihydrophenanthrene (54), 4,5-dihydroxy-2,3-dimethoxy-9,10-dih-ydrophenanthrene (56), 4,7-dihydroxy-2,3,6-trimethoxy-9,10-dihydrophenanthrene (57), 2,5-dihydr-oxy-3,4-dimethoxy-9,10-dihydrophenanthrene (59), 2,7-dihydroxy-3,4,6-trimethoxy-9,10-dihydrop-henanthrene (60), ephemeranthol A (63), denbinobin B (76)	[81,82]
47	*D. stuposum*	2,5-dihydroxy-4,9-dimethoxyphenanthrene (26), loddigesiinol A (27)	[83]
48	*D. thyrsiflorum*	moscatin (1), denthyrsinin (9), 2,6-dihydroxy-1,5,7-trimethoxyphenanthrene (24), hircinol (53), densi-florol B (68), denthyrsinol (89), denthyrsinone (88)	[84,85,86]
49	*D. trigonopus*	moscatin (1), hircinol (53)	[87]
50	*D. venustum*	flavanthrinin (2), lusianthridin (32), densiflorol B (68), phoyunnanin C (87), phoyunnanin E (94)	[88]
51	*D. wardianum*	moscatin (1), loddigesiinol A (27), coelonin (55), denbinobin (71), 3-ethoxy-5-hydroxy-7-methoxy-1,4-phenanthrenequinone (72), 9,10-dihydrodenbinobin (77)	[89,90]
52	*D. williamsonii*	-	-

**Table 2 molecules-27-07233-t002:** Bibenzyls isolated in *Dendrobium* species.

NO.	*Dendrobium* Species	Bibenzyls	Ref.
1	*D. amoenum*	amoenylin (125), isoamoenylin (127), 3,3′,4-trihydroxybibenzyl (134), moscatilin (148)	[6]
2	*D. aphyllum*	tristin (130), dihydroresveratrol (131), moscatilin(148), gigantol (150), batatasin III (151), 4,4′-dihy-droxy-3,5-dimethoxybibenzyl (157), aphyllal C (191), aphyllal D (193), aphyllal E (194), aphyllone B (214), trigonopol B (216)	[8,9]
3	*D. bellatulum*	aloifol I (128), 3,3′-dihydroxy-4,5-dimethoxybiphezyl (132), dendrosinen B (144), batatasin III (151), 4′,5-dihydroxy-3,3′-dimethoxybiphezyl (166), dendrosinen D (217)	[10]
4	*D. brymerianum*	tristin (130), moscatilin (148), gigantol (150)	[11]
5	*D. candidum*	3,4’-dihydroxy-5-methoxybibenzyl (135), 3,4-dihydroxy-4′,5-dimethoybibenzyl (140), dendrocandin E (145), moscatilin (148), chrysotoxine (149), gigantol (150), 4,4′-dihydroxy-3,5-dimethoxybibenzyl (157), chrysotobibenzyl (172), 3-O-methylgigantol (176), erianin(184), dendrocanol (188), dendro-bibenzy (190), dendrocandin A (195), dendrocandin C (198), dendrocandin D (199), dendrocandin F (218), dendrocandin G (219), dendrocandin J (220), dendrocandin K (221), dendrocandin B (224), dendrocandin I (225), dendrocandin H (226), dendrocandin L (227), dendrocandin N (229), dend-rocandin O (230), dendrocandin M (232), dendrocandin P (233), dendrocandin Q (234), dendromoni-liside E (235)	[13,91,92,93,94,95]
6	*D. capillipes*	moscatilin (148), chrysotoxine (149), gigantol (150), chrysotobibenzyl (172), crepidatin (185)	[96]
7	*D. catenatum*	dihydroresveratrol (131), 3,4’-dihydroxy-5-methoxybibenzyl (135), 3,4′,5-trihydroxy-3′-methoxy-bibenzyl (167), dendrocandin V (228), dendrocandin N (229), dendrocandin U (231), dendrocandin W (236), dendrofindlaphenol B (237)	[97]
8	*D. christyanum*	aloifol I (128), dendrosinen B (144), moscatilin (148), gigantol (150), batatasin III (151)	[15]
9	*D. chrysotoxum*	batatasin (129), tristin (130), moscatilin (148), chrysotoxine (149), gigantol (150), batatasin III (151), 4’,5-dihydroxy-3,3’-dimethoxybiphezyl(166), chrysotobibenzyl (172), erianin (184), trigonopol B (216), 3,3′,4′,5-tetramethoxybibenzyl-4-O-β-_D_-glucopyranoside (240), 3,4,4′,5-tetramethoxybibenzyl-3′-O-β-_D_-glucopyranoside (230)	[16,17,19,20,21,98]
10	*D. chrysanthum*	tristin (130), moscatilin (148), chrysotoxine (149), gigantol (150), batatasin III (151), chrysotobibenzyl (172), crepidatin (185)	[22,24,26]
11	*D. crepidatum*	moscatilin (148), 4′-hydroxy-3,4,5,3′-tetramethoxybibenzyl (168), erianin (184), crepidatin (185), crepidatuol A (242), crepidatuol B (243)	[27,99,100]
12	*D. crystallinum*	dihydroresveratrol (131), 3,4’-dihydroxy-5-methoxybibenzyl (135), gigantol (145), batatasin III (151), 4,4′-dihydroxy-3,5-dimethoxybibenzyl (157), 3,4′,5-trihydroxy-3′-methoxybibenzyl (167), 4′-hydroxy-3,3′,5-trimethoxybibenzyl (169), 3,5′-dihydroxy-3′,4-dimethoxybiphezyl (180), 5′-hydroxy-3,3′,4-trimethoxybibenzyl (186), dencryol A (238), dencryol B (239)	[101,102,103,104]
13	*D. denneanum*	moscatilin (148), gigantol (150), crepidatin (185)	[31]
14	*D. densiflorum*	tristin (130), moscatilin (148), gigantol (150), densiflorol A (192)	[33]
15	*D. devonianum*	tristin (130), 3-hydroxy-4′,5-dimethoxybibenzyl (136), 3,4′,5-trihydroxybibenzyl (154)	[35]
16	*D. draconis*	gigantol (150), batatasin III (151)	[36]
17	*D. ellipsophyllum*	moscatilin (148), 4,4′-dihydroxy-3,5-dimethoxybibenzyl (157), 4,5,4′-trihydroxy-3,3′-dimethoxybibe-nzyl (161)	[37]
18	*D. fimbriatum*	tristin (130), moscatilin (148), gigantol (150), batatasin III (151), 3,4’-dihydroxy-3′,4,5-trimethoxy-bibenzyl (170), crepidatin (185), 4,4′-dihydroxy-3,3′,5,α-tetramethoxybibenzyl (200), fimbriadimerbi-benzyl B (246), fimbriadimerbibenzyl C (247), fimbriadimerbibenzyl D (248), fimbriadimerbibenzyl F (249), fimbriadimerbibenzyl E (250), fimbriadimerbibenzyl G (251), gigantol-5-O-β-_D_-glucopyrano-side (253), trisin-5-O-β-_D_-glucopyranoside (254), fimbriadimerbibenzyl A (258)	[38,40]
19	*D. findlayanum*	3,4’-dihydroxy-5-methoxybibenzyl (135), 3,3’-dihydroxy-5-methoxybibenzyl (137), 3,4,4′-trihydroxy-5-methoxybibenzy (141), 3,4-dihydroxy-3’,4’,5-trimethoxybibenzyl (142), 4,4′-dihydroxy-3,5-dimeth-oxybibenzyl (157), 3′,4-dihydroxy-3,5-dimethoxybiphezyl (158), 4,4′-dihydroxy-3,5,3′-trimethoxy-bibenzyl (171), 3,3′-dihydroxy-4,5′-dimethoxybiphezyl (187), (R)-3,α-dihydroxy-4,4ʹ,5-trimethoxy-bibenzyl (207), dendrofindlaphenol B (237), dendrofindlaphenol A (255), dendrofindlaphenol C (256), 6″-de-O-methyldendrofindlaphenol A (257)	[105,106]
20	*D. formosum*	moscatilin (148), gigantol (150), batatasin III (151)	[41]
21	*D. gratiosissimum*	amoenylin (125), tristin (130), 3,4’-dihydroxy-5-methoxybibenzyl (135), 3-methylgigantol (138), 3,4-dihydroxy-4′,5-dimethoxybibenzyl (140), 3,4-dihydroxy-3’,4’,5-trimethoxybibenzyl (142), 3,4,4′-trihydroxy-5,3′-dimethoxybibenzyl (143), moniliformine (146), moscatilin (148), chrysotoxine (149), gigantol (150), batatasin III (151), 3,4′,5-trihydroxybibenzyl (154), aloifol (155), gigantol tetramethyl ether (156), 4,4′-dihydroxy-3,5-dimethoxybibenzyl (157), 4-hydroxy-3,5,4′-trimethoxybibenzyl (159), 3,4’-dihydroxy-3′,4,5-trimethoxybibenzyl (170), dendrogratiol A (178), dencryol A (238), dencryol B (239)	[107,28]
22	*D. huoshanense*	3,4’-dihydroxy-5-methoxybibenzyl (135), 3-hydroxy-4′,5-dimethoxybibenzyl (136), 3-hydroxy-5-methoxybibenzyl (139), batatasin III (151), 5,4′-dihydroxy-3-methoxybibenzyl (152), 4,4′-dihydroxy-3,5-dimethoxybibenzyl (157), 3’,4-dihydroxy-3,5′-dimethoxybibenzyl (189), dendrocandin B (224), dendrocandin U (231)	[108,109,110]
23	*D. hancockii*	3,4′,5-trihydroxy-3′-methoxybibenzyl (167), 4,4′-dihydroxy-3,5,3′-trimethoxybibenzyl (171), 3′,4-dihydroxy-3,5′-dimethoxybibenzyl (189), 3,α-dihydroxy-4,5,3′-trimethoxybibenzyl (202), crepidatuol B (243)	[42]
24	*D. harveyanum*	3,4-dihydroxy-4′,5-dimethoxybibenzyl (140), dendrofalconerol A (222), dendrocandin B (224), dendr-ofalconerol B (263)	[111]
25	*D. heterocarpum*	3,4’-dihydroxy-5-methoxybibenzyl (135), 3-hydroxy-4′,5-dimethoxybibenzyl (136), 3,4-dihydroxy-4′,5-dimethoxybibenzyl (140), moscatilin (148), gigantol (150), batatasin III (151), 3-O-methylgigantol (176), densiflorol A (192), dendrocandin A (195), (S)-3,4,α-trihydroxy-4′,5-dimethoxybibenzyl (196), dendrocandin F (218), dendrocandin I (225)	[43]
26	*D. hercoglossum*	3′,4-dihydroxy-3,5-dimethoxybiphezyl (158), 4,4′-dihydroxy-3,5,3′-trimethoxybibenzyl (171), 4,5-dihydroxy-3,3′,α-trimethoxybibenzyl (201), 3,4,α-trihydroxy-5,3′-dimethoxybibenzyl (203), 4,α-dihydroxy-3,5,3′-trimethoxybibenzyl (207)	[112]
27	*D. hongdie*	tristin (130), gigantol (150), batatasin III (151)	[44]
28	*D. hainanense*	-	-
29	*D. infundibulum*	aloifol I (128), 3,3′-dihydroxy-4,5-dimethoxybiphezyl (132), dendrosinen B (144), moscatilin (148), batatasin III (151), 5,4′-dihydroxy-3,4,3′-trimethoxybibenzyl (179)	[47]
30	*D. lindleyi*	tristin (130), 3,3′,5-trihydroxybibenzyl (133), 3-dendrobin A (147), moscatilin (148), chrysotoxine (149), gigantol (150), batatasin III (151), 4,5-dihydroxy-3,3′,4′-trimethoxybibenzyl (162), thunalbene (206)	[48,49]
31	*D. loddigesii*	tristin (130), 3,3′,5-trihydroxybibenzyl (133), moscatilin (148), gigantol (150), batatasin III (151), 4,4′-dihydroxy-3,5-dimethoxybibenzyl (157), 4,5,4′-trihydroxy-3,3′-dimethoxybibenzyl (161), 4′,5-dihy-droxy-3,3′-dimethoxybiphezyl (166), crepidatin (185), aphyllal C (191), densiflorol A (192), loddigesii-nol C (205), (R)-4,5,4ʹ-trihydroxy-3,3ʹ,α-trimethoxybibenzyl (208), loddigesiinol D (215), trigonopol B (216), crepidatuol B (243)	[50,51,53,54,113]
32	*D. longicornu*	aloifol I (128), batatasin (129), tristin (130), 3,3′-dihydroxy-4,5-dimethoxybiphezyl (132), 3,3′,4-trihy-droxybibenzyl (134), moscatilin (148), gigantol (150), batatasin III (151), 3,4’-dihydroxy-3′,4,5-trimeth-oxybibenzyl (170), cannabistilbene II (181), longicornuol A (244), trigonopol A (245)	[54,55,114]
33	*D. nobile*	tristin (130), 3,3′,5-trihydroxybibenzyl (133), 3-hydroxy-5-methoxybibenzyl (139), dendrobin A (147), moscatilin (148), chrysotoxine (149), gigantol (150), batatasin III (151), 3′,4-dihydroxy-3,5-dimethoxy-biphezyl (158), 4,5-dihydroxy-3,3′-dimethoxybibenzy (163), 5,3’-dihydroxy-3-methoxybibenzyl (164), 4′,5-dihydroxy-3,3′-dimethoxybiphezyl (166), 4′-hydroxy-3,3′,5-trimethoxybibenzyl (169), 4,4′-dihy-droxy-3,5,3′-trimethoxybibenzyl (171), chrysotobibenzyl (172), 3-O-methylgigantol (176), 3,4′-dihy-droxy-5,5′-dimethoxydihydrostilbene (182), crepidatin (185), nobilin D (197), 3′,4-dihydroxy-3,5′-dimethoxybibenzyl (189), 4,5-dihydroxy-3,3′,α-trimethoxybibenzyl (201), 4,α-dihydroxy-3,5,3′-trim-ethoxybibenzyl (204), nobilin B (209), nobilin A (212), nobilin C (213), nobilin E (223), dendronophen-ol A (252),dendronbibisline D (259), didendronbiline A (260), dendronbiline B (261), dendronbibisline C (262), dendronophenol B (266)	[56,57,59,60,115,116,117,118,119]
34	*D. moniliforme*	aloifol I (128), 3,3′-dihydroxy-4,5-dimethoxybiphezyl (132), moscatilin (148), gigantol (150), 3,4’-dihydroxy-3′,4,5-trimethoxybibenzyl (170), dendromoniliside E (235), longicornuol A (244)	[120,66]
35	*D. officinale*	amoenylin (125), tristin (130), 3,3′-dihydroxy-4,5-dimethoxybiphezyl (132), 3,4’-dihydroxy-5-methox-ybibenzyl (135), 3-hydroxy-4′,5-dimethoxybibenzyl (136), 3,4-dihydroxy-4′,5-dimethoxybibenzyl (140), 3,4,4′-trihydroxy-5-methoxybibenzy (141), dendrosinen B (144), moscatilin (148), gigantol (150), 4,4′-dihydroxy-3,5-dimethoxybibenzyl (157), 3,4′,5-trihydroxy-3′-methoxybibenzyl (167), 3-O-methylgigantol (176), 3,4′-dihydroxy-4,5-dimethoxybibenzyl (183), erianin(184), densiflorol A (192), (S)-3,4,α-trihydroxy-4′,5-dimethoxybibenzyl (196), trigonopol B (216), dendrocandin B (224), dendro-candin N (229), dendrocandin U (231), 6″-de-O-methyldendrofindlaphenol A (257), dendrocandin X (265), denofficin (267)	[67,68,121,122,123]
36	*D. pachyglossum*	Moscatilin (148), gigantol (150), 4,5,4′-trihydroxy-3,3′-dimethoxybibenzyl (161)	[124]
37	*D. palpebrae*	moscatilin (148), gigantol (150), 4,5,4′-trihydroxy-3,3′-dimethoxybibenzyl (161)	[70]
38	*D. parishii*	dendrocandin E (145), 4,3′,4′-trihydroxy-3,5-dimethoxybibenzyl (160), 4,5,4′-trihydroxy-3,3′-dimeth-oxybibenzyl (161)	[71]
39	*D. plicatile*	moscatilin (148), batatasin III (151), 5,3’-dihydroxy-3-methoxybibenzyl (164), 3’-hydroxy-3,4,4’,5-tetramethoxybibenzyl (174), 3-O-methylgigantol (176), 3,3’,4’-trimethoxy-5-hydroxybibenzyl (177)	[72,73]
40	*D. polyanthum*	moscatilin (148), gigantol (150), batatasin III (151)	[75]
41	*D. pulchellum*	moscatilin (138), chrysotoxine (139), chrysotobibenzyl (162), crepidatin (175)	[76]
42	*D. signatum*	4′,5′-dihydroxy-3′,4-dimethoxybiphezyl (126), 4,4′-dihydroxy-3,5-dimethoxybibenzyl (157), dendro-falconerol A (222), dendrocandin B (224), dendrocandin I (225), 6″-de-O-methyldendrofindlaphenol A (257)	[77,78]
43	*D. scabrilingue*	aloifol I (128), gigantol (150), batatasin III (151), dendroscabrol B (264)	[79]
44	*D. secundum*	moscatilin (148), 4,5,4′-trihydroxy-3,3′-dimethoxybibenzyl (161), 5-hydroxy-3,4,3′,4′,5′-pentamethox-ybibenzyl (173), brittonin A (175)	[96,125]
45	*D. senile*	aloifol I (128), moscatilin (148)	[80]
46	*D. sinense*	aloifol I (128), dendrosinen B (144), chrysotoxine (149), 5,3′-dihydroxy-3,4-dimethoxybibenzyl (165), 5,4′-dihydroxy-3,4,3′-trimethoxybibenzyl (179), dendrosinen A (210), dendrosinen C (211), dendrosi-nen D (217), longicornuol A (244), trigonopol A (245)	[82,126]
47	*D. stuposum*	4,4′-dihydroxy-3,5,3′-trimethoxybibenzyl (171)	[83]
48	*D. thyrsiflorum*	tristin (130), moscatilin (148), gigantol (150), batatasin III (151), 4′,5-dihydroxy-3,3′-dimethoxy-biphezyl (166), 3,4′,5-trihydroxy-3′-methoxybibenzyl (167), 4,4′-dihydroxy-3,5,3′-trimethoxybibenzyl (171), erianin (184)	[84,86,127]
49	*D. trigonopus*	tristin (130), moscatilin (148), gigantol (150), trigonopol B (216), trigonopol A (245)	[87,128]
50	*D. venustum*	gigantol (150), batatasin III (151)	[88]
51	*D. wardianum*	3,3′-dihydroxy-4,5-dimethoxybiphezyl (132), 3,4-dihydroxy-4′,5-dimethoxybibenzyl (140), moscatili-n (148), gigantol (150), 5-hydroxy-3,4′-dimethoxybibenzyl (153), 4-hydroxy-3,5,4′-trimethoxybibenzyl (159), 4′-hydroxy-3,3′,5-trimethoxybibenzyl (169), dendrocandin A (195), dendrocandin V (228), dendrocandin U (231)	[89,90]
52	*D. williamsonii*	amoenylin (125), aloifol I (128), 3,3′-dihydroxy-4,5-dimethoxybiphezyl (132), moniliformine (146), moscatilin (148), 4,4′-dihydroxy-3,5-dimethoxybibenzyl (157), dendrofindlaphenol A (256)	[129]

**Table 4 molecules-27-07233-t004:** Anti-inflammatory activity of stilbene compounds isolated from *Dendrobium* genus.

NO.	Compound	*Dendrobium* Species	NO Inhibition (%)	IC_50_ (µM)	Ref.
1	**1**	*D. aphyllum*	32.48 (25 μM)	-	[9]
2	**17**	*D. nobile*	-	20.4 ± 0.8	[59]
3	**20**	*D. nobile*	-	28.9 ± 0.6	[59]
4	**22**	*D. nobile*	-	35.7 ± 0.6	[59]
5	**27**	*D. loddigesii*	-	2.6	[52]
6	**32**	*D. nobile*	-	9.6 ± 0.3	[59]
7	**37**	*D. denneanum*	90 ± 7 (50 μM)	3.1	[29]
8	**38**	*D. denneanum*	86 ± 2 (50 μM)	4.2	[29]
9	**39**	*D. denneanum*	58 ± 8 (50 μM)	-	[29]
10	**41**	*D. loddigesii*	-	29.1	[52]
11	**43**	*D. loddigesii*	-	8.6	[51]
12	**49**	*D. nobile*	-	34.1 ± 0.9	[59]
13	**52**	*D. nobile*	-	17.6 ± 0.4	[59]
14	**53**	*D. nobile*	-	26.4 ± 0.2	[59]
15	**55**	*D. nobile*	-	10.2 ± 0.2	[59]
16	**57**	*D. nobile*	-	19.5 ± 0.4	[59]
17	**63**	*D. nobile*	-	12.0 ± 0.3	[59]
18	**79**	*D. aphyllum*	12.21 (25 μM)	-	[9]
19	**98**	*D. loddigesii*	-	9.9	[51]
20	**99**	*D. loddigesii*	-	10.9	[51]
21	**103**	*D. denneanum*	92 ± 2 (50 μM)	4.6	[29]
22	**104**	*D. denneanum*	76 ± 4 (50 μM)	16.9	[29]
23	**105**	*D. denneanum*	62 ± 1 (50 μM)	41.5	[29]
24	**106**	*D. denneanum*	68 ± 2 (50 μM)	-	[29]
25	**107**	*D. denneanum*	92 ± 5 (50 μM)	0.7	[29]
26	**115**	*D. loddigesii*	-	7.5	[51]
27	**116**	*D. loddigesii*	-	14.6	[51]
28	**130**	*D. aphyllum*	27.23 (25 μM)	-	[9]
29	**131**	*D. aphyllum*	25.82 (25 μM)	-	[9]
30	**133**	*D. loddigesii*	-	13.1	[51]
31	**148**	*D. nobile*	-	27.6 ± 0.5	[59]
32	**150**	*D. nobile*	-	32.9	[115]
33	**151**	*D. loddigesii*	-	21.9	[51]
34	**157**	*D. loddigesii*	-	49.3	[51]
35	**172**	*D. nobile*	-	48.2	[115]
36	**185**	*D. crepidatum*	-	3.04 ± 1.15	[100]
37	**191**	*D. aphyllum*	22.07 (25 μM)	-	[9]
38	**193**	*D. aphyllum*	14.96 (25 μM)	-	[9]
39	**197**	*D. nobile*	-	15.3	[115]
40	**209**	*D. crepidatum*	-	26.64 ± 0.51	[100]
41	**214**	*D. loddigesii*	-	69.7	[52]
42	**215**	*D. aphyllum*	19.72 (25 μM)	-	[9]
43	**216**	*D. loddigesii*	-	26.3	[51]
44	**223**	*D. nobile*	-	19.2	[115]
45	**257**	*D. findlayanum*	-	21.4 ± 1.0	[106]

**Table 5 molecules-27-07233-t005:** IC_50_ values of stilbene compounds isolated from *Dendrobium* genus for α-glucosidase and pancreatic lipase inhibitory activities.

No.	Compound	*Dendrobium* Species	α-Glucosidase (µM)	Pancreatic Lipase (µM)	Ref.
1	**1**	*D. senile*	-	57.60 ± 3.30	[80]
2	**14**	*D. formosum*	189.78 ± 1.11	154.61 ± 8.58	[41]
3	**18**	*D. scabrilingue*	96.2 ± 12.0	-	[79]
4	**26**	*D. senile*	-	58.60 ± 3.40	[80]
5	**32**	*D. scabrilingue*	112.9 ± 5.3	-	[79]
6	**33**	*D. christyanum,*	133.11 ± 10.82	-	[15]
7	**43**	*D. loddigesii*	119.2	-	[51]
8	**55**	*D. scabrilingue*	131.4 ± 6.6	-	[79]
9	**73**	*D. formosum*	126.88 ± 0.66	69.45 ± 10.14	[41]
10	**98**	*D. loddigesii*	52.8	-	[51]
11	**99**	*D. loddigesii*	39.6	-	[51]
12	**115**	*D. loddigesii*	5.5	-	[51]
13	**116**	*D. loddigesii*	5.8	-	[51]
14	**123**	*D. loddigesii*	16.7	-	[53]
15	**124**	*D. loddigesii*	10.9	-	[53]
16	**131**	*D. catenatum*	36.05 ± 0.67	-	[97]
17	**133**	*D. loddigesii*	31.8	-	[51]
18	**135**	*D. catenatum*	159.59 ± 0.86	-	[97]
19	**140**	*D. harveyanum*	32.0 ± 1.5	NA	[111]
20	**141**	*D. officinale*	403.4	-	[121]
21	**144**	*D. infundibulum*	213.9 ± 2.4	295.01 ± 37.90	[47]
22	**150**	*D. christyanum,*	79.87 ± 14.20	-	[15]
23	**151**	*D. infundibulum*	148.8 ± 8.4	NA	[47]
24	**157**	*D. loddigesii*	94.2	-	[51]
25	**222**	*D. harveyanum*	71.6 ± 5.5	6.60 ± 0.10	[111]
26	**231**	*D. officinale*	9.46	-	[121]
27	**235**	*D. harveyanum*	51.2 ± 3.4	4.90 ± 0.09	[111]
28	**243**	*D. loddigesii*	18.9	-	[53]
29	**253**	*D. scabrilingue*	9.4 ± 0.7	-	[75]

**Table 6 molecules-27-07233-t006:** The antiproliferative activity of the stilbenes against five cancer cells (IC_50_ µM).

Compound	MCF-7	HL-60	A549	SMMC-7721	SW480	Ref.
**1**	23.75 ± 0.82	NA	16.29 ± 0.25	NA	18.97 ± 1.04	[89]
**71**	13.13 ± 0.47	3.08 ± 0.12	19.68 ± 1.12	NA	16.81 ± 0.13	[89]
**77**	3.63 ± 0.03	2.33 ± 0.12	14.79 ± 0.64	14.84 ± 0.41	6.66 ± 0.71	[89]
**148**	6.20	2.18	5.73	4.07	6.88	[38]
**170**	37.12	24.32	>40	>40	>40	[38]
**228**	38.48 ± 1.16	NA	NA	NA	NA	[89]
**246**	12.06	15.79	15.55	16.63	18.19	[38]
**247**	24.84	22.33	>40	>40	>40	[38]
**249**	13.67	19.08	16.64	17.47	18.83	[38]
**250**	5.85	16.91	16.49	18.89	21.23	[38]
**251**	7.75	16.91	16.49	18.89	21.23	[38]
**255**	4.7 ± 0.7	2.3 ± 0.2	4.4 ± 0.5	5.1 ± 0.4	5.3 ± 0.8	[106]
**257**	19.4 ± 2.7	23.5 ± 2.0	20.8 ± 1.6	34.4 ± 2.3	>40	[106]
**258**	12.67	16.13	16.34	18.27	17.15	[106]

**Table 7 molecules-27-07233-t007:** The IC_50_ of stilbenes inhibiting cancer cells (IC_50_ µM).

NO.	Compound	*Dendrobium* Species	Cancer cell	IC_50_ (µM)	Ref.
1	**3**	*D. devonianum*	HT-29	30.30	[34]
2	**7**	*D. plicatile*	HepG2	6.12	[73]
			MDA-MB231	12.46~19.61	
			A549	11.68~43.88	
3	**9**	*D. thyrsiflorum*	Hela	2.7	[86]
			K-562	2.3	
			MCF-7	4.8	
4	**14**	*D. officinale*	HL-60	18.95 ± 0.70	[123]
			THP-1	11.51 ± 0.12	
5	**15**	*D. plicatile*	HepG2	4.20	[73]
			A549	9.12	
6	**26**	*D. chrysotoxum*	K562	45.64	[17]
			HL-60	1~10	
			A549	8.65	
			BEL-7402	1.79	
			SGC-7901	2.89	
		*D. plicatile*	HepG2	11.27~38.68	[73]
7	**32**	*D. brymerianum*	H460	65.0	[11]
		*D. nobile*	A549	9.8	[58]
			HL-6	7.7	
			SK-OV-3	9.4	
		*D. plicatile*	HepG2	8.70	[73]
			MDA-MB231	8.04	
8	**43**	*D. officinale*	HL-60	29.53 ± 0.22	[123]
			THP-1	26.53 ± 0.58	
9	**47**	*D. officinale*	HL-60	11.96 ± 0.58	[123]
			THP-1	8.92 ± 0.67	
10	**53**	*D. plicatile*	HepG2	11.27~38.68	[73]
			A549	11.27~38.68	
		*D. thyrsiflorum*	K-562	6.3	[86]
11	**55**	*D. plicatile*	HepG2	9.35	[73]
			MDA-MB231	5.36	
			A549	5.79	
12	**58**	*D. plicatile*	HepG2	8.40	[73]
			MDA-MB231	12.46~19.61	
			A549	11.68~43.88	
13	**63**	*D. officinale*	HL-60	39.35 ± 1.58	[123]
			THP-1	36.34 ± 2.21	
		*D. plicatile*	HepG2	11.27~38.68	[73]
			MDA-MB231	12.46~19.61	
			A549	11.68~43.88	
14	**64**	*D. plicatile*	HepG2	11.27~38.68	[73]
15	**67**	*D. plicatile*	HepG2	8.84	[73]
			MDA-MB231	7.69	
			A549	8.92	
16	**71**	*D. nobile*	SK-OV-3	3.5	[58]
17	**88**	*D. thyrsiflorum*	Hela	9.9	[86]
			K-562	6.5	
			MCF-7	3.0	
18	**89**	*D. thyrsiflorum*	Hela	9.3	[86]
			K-562	1.6	
19	**108**	*D.denneanum*	SNU387	4.38	[30]
20	**117**	*D.denneanum*	SNU387	8.40	[30]
		*D. nobile*	HepG2	4.81 ± 0.04	[60]
21	**118**	*D.denneanum*	SNU387	11.21	[30]
		*D. nobile*	HepG2	19.47 ± 1.11	[60]
22	**119**	*D. officinale*	HL-60	35.32 ± 1.76	[123]
			THP-1	20.78 ± 1.80	
23	**120**	*D. officinale*	THP-1	45.32 ± 2.39	[123]
24	**125**	*D. gratiosissimum*	HepG2	10.15	[28]
			BGC823	23.12	
25	**128**	*D. sinense*	SGC-7901	12.8 ± 0.6	[126]
		*D. williamsonii*	HL-60	5.10	[129]
26	**130**	*D. officinale*	HL-60	29.06 ± 1.13	[123]
			THP-1	13.53 ± 1.10	
27	**132**	*D. williamsonii*	KB	195.0	[132]
			MCF-7	187.7	
28	**135**	*D. gratiosissimum*	HL-60	89.8	[107]
29	**137**	*D. findlayanum*	SHSY5Y	19.40 ± 1.95	[105]
			Hela	26.71 ± 2.26	
30	**139**	*D. gratiosissimum*	HL-60	45.6	[107]
31	**140**	*D. gratiosissimum*	HL-60	25.6	[107]
			U87-MG	24.24	[28]
			HepG2	35.19	
32	**142**	*D. findlayanum*	A172	12.26 ±1.48	[105]
33	**143**	*D. gratiosissimum*	HepG2	31.40	[28]
34	**146**	*D. williamsonii*	HL-60	10.69	[129]
35	**148**	*D. brymerianum*	H460	196.7	[11]
		*D. capillipes*	KB	2.2	[96]
			NCI-H187	10.5	
		*D. ellipsophyllum*	H292	226.09 ± 5.67	[37]
		*D. officinale*	Hela	16.8 ± 2.6	[67]
		*D. plicatile*	HepG2	11.27~38.68	[73]
		*D. pulchellum*	H23	33.41 ± 5.36	[76]
		*D. thyrsiflorum*	K-562	7.1	[86]
36	**149**	*D. capillipes*	KB	60.5	[96]
			NCI-H187	65.6	
		*D. pulchellum*	H23	198.38 ± 9.28	[76]
37	**150**	*D. brymerianum*	H460	23.4	[11]
		*D. capillipes*	KB	61.9	[96]
			NCI-H187	71.6	
			MCF-7	67.8	
		*D. gratiosissimum*	HL-60	10.6	[107]
		*D. officinale*	Hela	92.4 ± 6.4	[67]
			THP-1	23.34 ± 0.83	[123]
38	**151**	*D. plicatile*	HepG2	11.27~38.68	[73]
			MDA-MB231	12.46~19.61	
			A549	11.68~43.88	
39	**157**	*D. ellipsophyllum*	H292	197.74 ± 0.78	[37]
		*D. findlayanum*	A172	13.14 ± 2.31	[105]
			SHSY5Y	4.05 ± 0.76	
			Hela	5.99 ± 1.15	
		*D. gratiosissimum*	HCT116	33.66	[28]
			HepG2	15.25	
			BGC823	22.07	
			PC9	45.36	
40	**158**	*D. findlayanum*	SHSY5Y	13.37 ± 1.12	[105]
41	**159**	*D. gratiosissimum*	HepG2	47.44	[28]
			PC9	30.63	
42	**161**	*D. ellipsophyllum*	H292	96.56 ± 0.22	[37]
		*D. capillipes*	KB	48.3	[96]
			NCI-H187	63.8	
			MCF-7	62.6	
43	**165**	*D. sinense*	SGC-7901	7.8 ± 0.05	[126]
			BEL-7402	11.7 ± 0.5	
			K562	15.7 ± 0.2	
44	**171**	*D. findlayanum*	A172	3.77 ± 0.60	[105]
			SHSY5Y	1.65 ± 0.16	
			Hela	2.22 ± 0.25	
45	**172**	*D. capillipes*	KB	132.4	[96]
			NCI-H187	123.7	
		*D. pulchellum*	H23	252.14 ± 12.21	[76]
46	**173**	*D. capillipes*	NCI-H187	87.8	[96]
47	**176**	*D. plicatile*	HepG2	11.27~38.68	[73]
			MDA-MB231	12.46~19.61	
			A549	11.68~43.88	
		*D. sinense*	SGC-7901	16.7 ± 0.4	[126]
48	**184**	*D. thyrsiflorum*	K-562	0.0014	[86]
49	**185**	*D. capillipes*	KB	14.4	[96]
			NCI-H187	13.7	
		*D. pulchellum*	H23	157.77 ± 11.21	[76]
50	**207**	*D. findlayanum*	A172	34.69 ± 2.95	[105]
			SHSY5Y	10.63 ± 1.05	
			Hela	15.74 ± 1.87	
51	**224**	*D. officinale*	Hela	91.1 ± 11.2	[67]
52	**231**	*D. officinale*	Hela	41.5 ± 2.4	[67]
53	**232**	*D. sinense*	BEL-7402	10.0 ± 0.4	[126]
			K562	10.3 ± 0.1	
54	**238**	*D. gratiosissimum*	HL-60	2.1	[107]
55	**239**	*D. gratiosissimum*	HL-60	6.4	[107]
56	**259**	*D. nobile*	HepG2	1.25 ± 0.06	[60]
57	**262**	*D. nobile*	HepG2	11.99 ± 1.02	[60]
58	**267**	*D. officinale*	Hela	20.2 ± 1.3	[67]

## Data Availability

Not Applicable.

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
