# Peer review of "Recent Research Progress on Natural Stilbenes in Dendrobium Species"

_molecules, 2022, doi:10.3390/molecules27217233_

Round 1

Reviewer 1 Report

The review article entitled “Recent research progress on natural stilbenes in Dendrobium species” describes the sources, pharmacological activities and mechanism of actions of stilbene compounds in 50 species of Dendrobium species, which have been used in traditional Chinese medicine.  The review article is written well, and I believe it will be valuable document for both researchers and reader having interest in Dendrobium species especially on natural stilbenes. I would like to recommend it for publication after minor revision.

Minor comments –

1.    The quality of the manuscript will enhance if author able to describe the methodology used for literature search and the timeline.

2.    Even though more than 1500 Dendrobium species exist (lines 28-30), the article covers stilbenes reported from only 50 species (lines (50-52). Better to have explanation why 50 species were chosen in this review paper?

3.    The article only describe the positive biological activities of stilbenes. Is there any negative health effect of stilbenes reported or any toxicology data in literature? Encourage to add a separate section describing negative effect of stilbenes.  

4.    Topographical error IC50 should be IC50 and H2O2 should be H2O2 throughout the manuscript.

Author Response

Dear Editors and Reviewers:

  Thank you again for your letter and for the reviewer’s comments concerning our manuscript entitled “Recent research progress on natural stilbenes in Dendrobium species”. Those comments were all valuable and very helpful in revising and improving our paper, as well as being an excellent guide to our research. We have carefully studied the comments and have made corrections that we hope will meet with approval. Meanwhile, we have marked all revision with the function of “Track Changes” as required. The main revisions in the paper and the responses to the reviewer’s comments are as following:

Responds to the reviewer's comments:

Reviewer #1

comment 1. The quality of the manuscript will enhance if author able to describe the methodology used for literature search and the timeline.

Response 1: ​Following your valuable suggestions, we have added the database, keywords, and timeline used to collect the data in the introduction.

comment 2. Even though more than 1500 Dendrobium species exist (lines 28-30), the article covers stilbenes reported from only 50 species (lines (50-52). Better to have explanation why 50 species were chosen in this review paper?

Response 2: According to your comment, we have added the reason for choosing these Dendrobium species in the introduction.

comment 3. The article only describe the positive biological activities of stilbenes. Is there any negative health effect of stilbenes reported or any toxicology data in literature? Encourage to add a separate section describing negative effect of stilbenes.

Response 3: Through literature review, many Dendrobium species have been utilized as a traditional Chinese medicine (TCM) for thousands of years in China, but no obvious toxic side effects and adverse effects have been found so far. Furthermore, activity investigations of most of the stilbene components in Dendrobium have been limited to in vitro screening, and few systematic pharmacological studies are available in the literature due to the scarcity of samples. Although positive results have been obtained in most cell culture and animal studies, further human studies are needed to substantiate the beneficial and negative effects of those stilbene components.

comment 4. Topographical error IC50 should be IC50 and H2O2 should be H2O2 throughout the manuscript.

Response 4: I'm so sorry for our carelessness, we have modified it now.

We tried our best to improve the manuscript and made some changes in the manuscript. We appreciate for editors and reviewers’ warm work earnestly, and hope that the correction will meet with approval.

  Once again, special thanks to you for your good comments and suggestions.

Sincerely yours,

Jinyan Cai

Reviewer 2 Report

The manuscript needs substantial improvement. Unfortunately, the manuscript in its current form is not well-organized and poorly presented. Therefore, I recommend the authors' team consider the following points during the revision of the manuscript.

  1. The introduction section needs additional refinement. Please, expand the background of the topic and highlight the role of stilbenes as antiviral drugs, especially against HSV infections. This information can be extracted from the reference (DOI: 10.3390/v12020154).
  2. The authors should highlight information about the databases used for collecting/extracting the data (for example, Web of Science, Scopus, Google Scholar,..) and what keywords were used during the literature search along with the period of studies included in the review. This ensures that the paper covers all available recent and relevant studies. All these points could be highlighted, at least, in the introduction section.
  3. Analyzing and discussing the collected data critically. In other words, the reviewed studies and the acquired data should be deeply discussed and interpreted. Unfortunately, the manuscript in its current form looks like a hasty report. The authors should carefully consider this point. 
  4.  Drawing figures that display the reported mechanisms of action of the reviewed compounds. This will help the readers get directly into the point, especially the readers who prefer to look at display items without reading the corresponding text.
  5. It would be better to add a new section that provides information about the most effective reviewed compounds that show promising pharmacological actions against the reviewed targets in preclinical and clinical studies, and could be used as templates for further development, especially for medicinal chemists.
  6. It would be better to add a new section that discusses the current and future strategies for developing effective and safe stilbene-type compounds.
  7. The abstract and conclusion sections should be revised according to the newly added information.
  8. Finally, I recommend the authors' team double-check the whole manuscript for grammatical and typing errors by seeking the help of a native English speaker. 

Author Response

Dear Editors and Reviewers:

  Thank you again for your letter and for the reviewer’s comments concerning our manuscript entitled “Recent research progress on natural stilbenes in Dendrobium species”. Those comments were all valuable and very helpful in revising and improving our paper, as well as being an excellent guide to our research. We have carefully studied the comments and have made corrections that we hope will meet with approval. Meanwhile, we have marked all revision with the function of “Track Changes” as required. The main revisions in the paper and the responses to the reviewer’s comments are as following:

Responds to the reviewer's comments:

Reviewer #2

comment 1. The introduction section needs additional refinement. Please, expand the background of the topic and highlight the role of stilbenes as antiviral drugs, especially against HSV infections. This information can be extracted from the reference (DOI: 10.3390/v12020154).

Response 1: First of all, thank you very much for your comments. Unfortunately, after carefully reading your recommendation, I found only one stilbene, resveratrol, in the chemical composition of this article. In addition, resveratrol was not isolated from Dendrobium. We also searched again in different databases for antiviral effects of the stilbene components of Dendrobium. However, most of the current research on the stilbene component of Dendrobium is in vitro, and due to the small number of samples, there are few studies in the literature on its antiviral effects.​ We will strengthen our research on the antiviral effects of stilbene components in Dendrobium species in the future.

comment 2. The authors should highlight information about the databases used for collecting/extracting the data (for example, Web of Science, Scopus, Google Scholar,..) and what keywords were used during the literature search along with the period of studies included in the review. This ensures that the paper covers all available recent and relevant studies. All these points could be highlighted, at least, in the introduction section.

Response 2: Following your valuable suggestions, we have added the database, keywords, and timeline used to collect the data in the introduction.

comment 3. Analyzing and discussing the collected data critically. In other words, the reviewed studies and the acquired data should be deeply discussed and interpreted. Unfortunately, the manuscript in its current form looks like a hasty report. The authors should carefully consider this point.

Response 3: Following your valuable suggestions, we further analyzed and discussed the collected data.

comment 4. Drawing figures that display the reported mechanisms of action of the reviewed compounds. This will help the readers get directly into the point, especially the readers who prefer to look at display items without reading the corresponding text.

Response 4: According to your comment, we summarize the main pharmacological mechanisms of stilbene compounds in Dendrobium species and plot the figures.​

comment 5. It would be better to add a new section that provides information about the most effective reviewed compounds that show promising pharmacological actions against the reviewed targets in preclinical and clinical studies, and could be used as templates for further development, especially for medicinal chemists.

Response 5: At present, activity investigations on most stilbene compounds of Dendrobium are only limited to screening in vitro, and few systematic pharmacological, preclinical and clinical studies. ​In the section on future prospects, we selected a number of stilbene compounds with good activity and potential to be developed into drugs based on the references in the hope of providing a scientific basis for further pharmacological studies and new drug development of stilbene.​

comment 6. It would be better to add a new section that discusses the current and future strategies for developing effective and safe stilbene-type compounds.

Response 6: According to your comment, we have added a new section of future prospects in the article.

comment 7. The abstract and conclusion sections should be revised according to the newly added information.

Response 7: We have revised the abstract and conclusions in light of the new inform.

comment 8. Finally, I recommend the authors' team double-check the whole manuscript for grammatical and typing errors by seeking the help of a native English speaker.

Response 8: We have sought help to examine the grammar and spelling of the text, and have endeavoured to correct the grammar and language.​

We tried our best to improve the manuscript and made some changes in the manuscript. We appreciate for editors and reviewers’ warm work earnestly, and hope that the correction will meet with approval.

  Once again, special thanks to you for your good comments and suggestions.

Sincerely yours,

Jinyan Cai

Round 2

Reviewer 2 Report

The manuscript has been sufficiently improved.